# UAV-Based Forest Health Monitoring: A Systematic Review

Simon Ecke [1,2,*], Jan Dempewolf [1], Julian Frey [2], Andreas Schwaller [1], Ewald Endres [3], Hans-Joachim Klemmt [1], Dirk Tiede [4] and Thomas Seifert [2,5]

[1] Bavarian State Institute of Forestry, Hans Carl-von-Carlowitz-Platz 1, 85354 Freising, Germany; jan.dempewolf@lwf.bayern.de (J.D.); andreas.schwaller@lwf.bayern.de (A.S.); hans-joachim.klemmt@lwf.bayern.de (H.-J.K.)

[2] Chair of Forest Growth and Dendroecology, Faculty of Environment and Natural Resources, University of Freiburg, Tennenbacher Str. 4, 79106 Freiburg, Germany; julian.frey@iww.uni-freiburg.de (J.F.); thomas.seifert@iww.uni-freiburg.de (T.S.)

[3] Faculty of Forestry, University of Applied Sciences Weihenstephan-Triesdorf, 85354 Freising, Germany; ewald.endres@hswt.de

[4] Department of Geoinformatics—Z_GIS, University of Salzburg, Schillerstr. 30, 5020 Salzburg, Austria; dirk.tiede@plus.ac.at

[5] Department of Forest and Wood Science, Stellenbosch University, Private Bag X1, Matieland 7602, South Africa

* Correspondence: simon.ecke@iww.uni-freiburg.de

**Abstract:** In recent years, technological advances have led to the increasing use of unmanned aerial vehicles (UAVs) for forestry applications. One emerging field for drone application is forest health monitoring (FHM). Common approaches for FHM involve small-scale resource-extensive fieldwork combined with traditional remote sensing platforms. However, the highly dynamic nature of forests requires timely and repetitive data acquisition, often at very high spatial resolution, where conventional remote sensing techniques reach the limits of feasibility. UAVs have shown that they can meet the demands of flexible operation and high spatial resolution. This is also reflected in a rapidly growing number of publications using drones to study forest health. Only a few reviews exist which do not cover the whole research history of UAV-based FHM. Since a comprehensive review is becoming critical to identify research gaps, trends, and drawbacks, we offer a systematic analysis of 99 papers covering the last ten years of research related to UAV-based monitoring of forests threatened by biotic and abiotic stressors. Advances in drone technology are being rapidly adopted and put into practice, further improving the economical use of UAVs. Despite the many advantages of UAVs, such as their flexibility, relatively low costs, and the possibility to fly below cloud cover, we also identified some shortcomings: (1) multitemporal and long-term monitoring of forests is clearly underrepresented; (2) the rare use of hyperspectral and LiDAR sensors must drastically increase; (3) complementary data from other RS sources are not sufficiently being exploited; (4) a lack of standardized workflows poses a problem to ensure data uniformity; (5) complex machine learning algorithms and workflows obscure interpretability and hinders widespread adoption; (6) the data pipeline from acquisition to final analysis often relies on commercial software at the expense of open-source tools.

**Keywords:** unmanned aerial vehicle; stress detection; forest health monitoring; remote sensing; forestry; machine learning; multispectral; hyperspectral; structure from motion

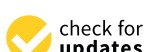



## 1. Introduction

Extensive ground surveys serve as a valuable pool for indicators of tree and forest health. Since these are extremely labor and cost intensive, their implementation is limited to the plot level. To obtain information at regional scales, individual plots must be aggregated. Detailed information at this intermediate level is often missing but is needed to extend monitoring to larger areas using remote sensing (RS) techniques [1,2]. A gap has formed

between in situ ground observations and traditional RS platforms. This review analyzes the literature related to the emerging field of UAV-based forest health monitoring (FHM) to evaluate its suitability to fill this gap.

### 1.1. UAV Remote Sensing: An Emerging Tool in Forest Health Monitoring

To assess the severity and the spatial extent of the consequences of tree stress, FHM is necessary. Timely knowledge of forest health decline might enable forest management to mitigate or reduce the damage to the forest (e.g., pest treatment, fire prevention) as well as to the ecosystem services attached to it (e.g., timber production, drinking water purification) [3,4].

Long-term in situ monitoring of forest health has been carried out already for several decades at local, regional, and global scales in which standardized forest health indicators have been included [5]. In the process, typically ground-truth information is collected by trained forestry staff that indicates an external state of health or damage. A common procedure is the estimation of the loss of needles and leaves, which is recorded in 5% intervals based on site-specific example trees [6,7]. The ICP Forest program is a European collaboration of 42 participating countries and provides a standardized framework for the assessment and monitoring of forests in Europe and beyond [8]. They act on two different intensity levels, namely, level-1 monitoring, which includes 5714 inventory plots (as of 2020) based on a systematic grid throughout Europe to assess temporal variations of the forest condition, and the level-2 monitoring, which investigates 623 plots (as of 2018) to identify cross-correlations between environmental factors and the responses of forests in selected ecosystems [9]. For a detailed overview of other national and international in situ FHM programs, please refer to [5].

A high degree of uncertainty is to be expected of in situ ground surveys as the quality depends on the experience and the subjective individual perception of the observers. Given that these are logistically complex, time-consuming, cost and labor-intensive ground surveys, only feasible at the plot-scale [10], it is not surprising that remote sensing (RS) has established itself as a part of FHM and its research and has made major contributions to this application field [5,11–14]. RS enables the acquisition of forest health indicators based on spectral or structural features derived from sensor data in an objective, quantitative, and repetitive manner at multiple spatial scales [12].

Unmanned aerial vehicles (UAVs), sometimes referred to as drones or unmanned aircraft (UA), are remotely piloted carrier platforms and are increasingly being used as RS tools in various forestry applications [15,16]. The terms unmanned aircraft system (UAS) and, less frequently, remotely piloted aircraft system (RPAS), considering the UAV as part of an entire operating system, including the equipment needed to control it remotely, have been widely adopted [17]. Important to note is that the Department of National Defence and Canadian Armed Forces (DND/CAF) has exchanged the term "unmanned air vehicle" to "uncrewed air vehicle" to ensure a gender neutral terminology [18]. Other nations should consider a similar approach. Following this example, we have adopted the terms "crewed" and "uncrewed" in the remainder of this review.

Satellite-based RS still dominates forest health research. The most frequently used data are publicly accessible imagery such as Landsat, MODIS, and Sentinel [19]. The large area coverage makes satellites valuable tools at multiple scales. Nevertheless, both the temporal and spatial resolution of satellite RS for monitoring forest ecosystems are sometimes insufficient and reach the limits of feasibility [20]. This is particularly the case when biotic or abiotic factors trigger responses that lead to rapidly changing processes in forests and thus detrimentally affect forest health [21,22]. In such cases, up-to-date and regularly acquired information becomes a key requirement [23]. Besides timeliness, a very high spatial resolution is also critical in sustainable forest management [24].

In theory, crewed aircraft can meet these requirements. In practice, though, safety issues, logistical limitations and very high costs make them viable only for relatively large areas [25]. It is precisely in this niche in forestry where drones are increasingly used and are therefore seen less as a competitor than as a complementary technology to existing

traditional Earth observation platforms [26]. While inferior to satellites and crewed RS platforms in terms of area coverage, they are unrivaled in spatial resolution, cost-efficiency, flexibility, and frequent revisit times, provided the area of interest is relatively small in size [27]. Common area coverage ranges from one hectare to several square kilometers in a single flight. The exact area coverage depends on the UAV type, propulsion technology (electricity/fuel), camera type, and other circumstances, such as rugged terrain (mountainous regions) or accessibility. A comparison of crewed aircraft and UAVs distributed by spatial resolution, costs per m² and area coverage can be seen in Figure 1. In addition, the regulations of UAV operation of the respective country need to be considered, which can limit the maximum allowed flight height above ground level and thus also may influence the possible area coverage.

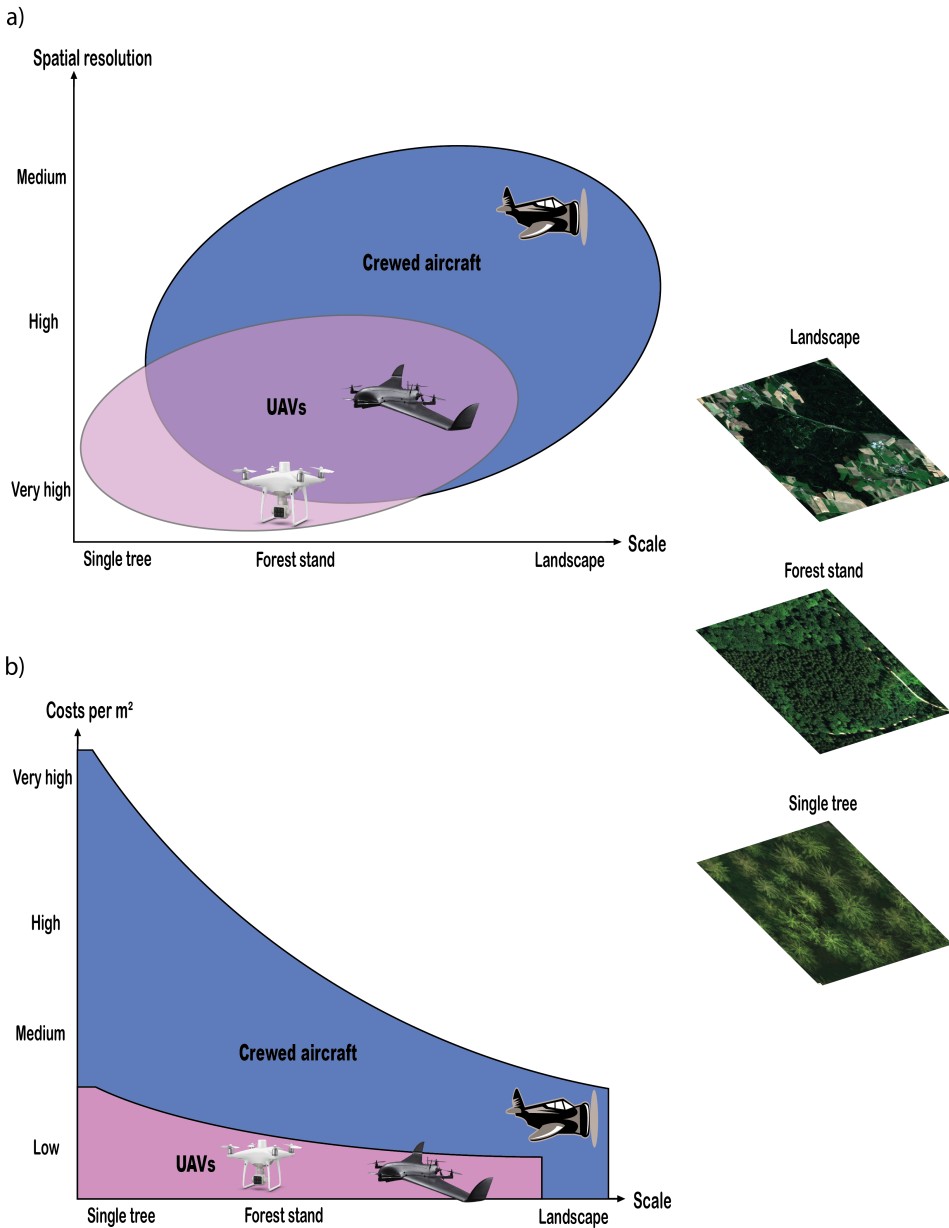

**Figure 1.** Abstraction of the distribution of potentially beneficial use cases of crewed aircraft and different UAV types used for remote sensing in forestry: (**a**) by scale and spatial resolution, (**b**) by scale and costs per m². Note: within the overlap area in the top graph, UAVs often have a significant cost advantage over crewed aircraft and are faster and more frequently deployable. Multirotor drones are easier to handle compared to fixed-wing aircraft, but are only suitable for relatively small areas.

### 1.2. UAV Types and Sensors for Forest Health Monitoring

UAVs can generally be controlled manually and perform predetermined flight missions autonomously. For a general UAV workflow for data acquisition see Figure 2.

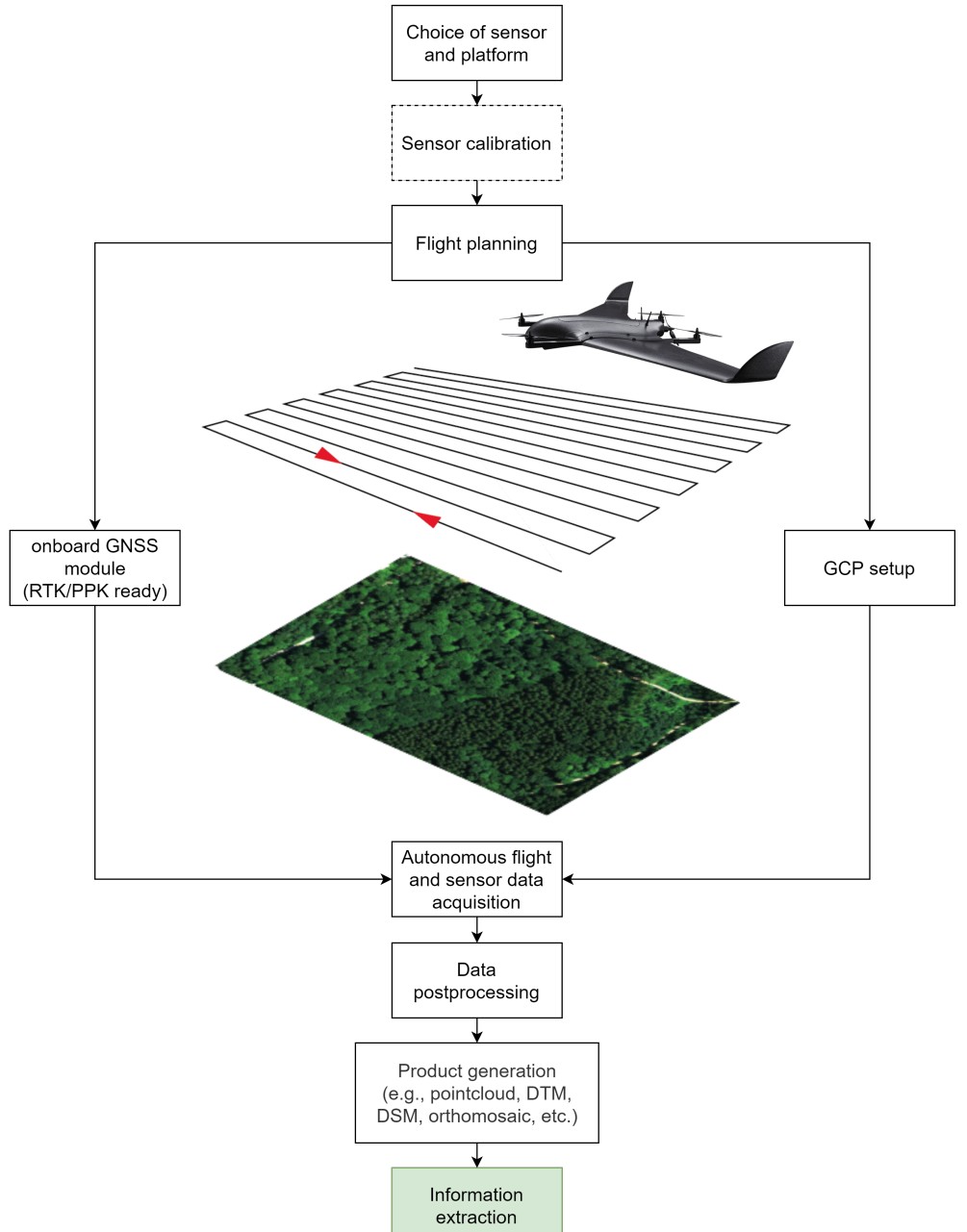

**Figure 2.** General workflow of UAV-based data acquisition and processing procedures.

The platforms themselves come in various sizes (small, mini, and micro) and shapes, with maximum take-off weights (MTOW) most likely below 30 kg [28] due to legal requirements that make operation above the MTOW complicated and cumbersome [29]. Flight times vary depending on weather conditions, payload, battery power or engine fuel, and UAV type and vary from a few minutes to several hours [30]. With typical ranges not greater than 10 km [17], the legally possible capacity can usually be exhausted, as flying beyond the visual line of sight (BVLOS) is not permitted in many countries. Fixed-wing, rotary-wing, and hybrid solutions represent the major platform types that have prevailed for recent forestry applications [16]. The former has the shape of an ordinary airplane with rigid wings and benefits from its aerodynamic shape to optimize cruise efficiency. This

allows it to maximize flight duration and, thus, larger areas can be covered [31]. The main drawback is that fixed-wing models require large and flat take-off and landing locations without higher obstacles [32], which can pose problems, especially in forested terrain. Moreover, operators must consider that this platform type needs a constant forward motion with a minimum speed to get sufficient air buoyancy. This could lead to motion blur, which adversely affects image quality [33].

Rotary-wing drones are most frequently embodied by multirotor models propelled by four, six, or up to eight rotors [17]. Single-rotor models (helicopters) are still being used in a few cases [34]. Multirotor UAVs are capable of vertical take-off and landing (VTOL) are much more maneuverable and responsive, allowing operators to launch even in locations with constrained space. They are easier to handle and are generally more versatile in their applications [35]. The ability to fly slowly around objects or hover in place opens up additional applications such as data capturing for fine three-dimensional modeling where detailed multiview side images are necessary [15,36,37]. However, all of this comes at the expense of higher energy consumption [38]. Hence, only small areas are feasible with a multirotor drone compared to its fixed-wing equivalent. Even if more expensive, hybrid VTOL fixed-wing solutions alleviate the problem of the need for large take-off and landing areas [30], which makes them particularly interesting for forestry applications. These platforms combine rigid wings and multiple rotors that enable VTOL operation and typically transition from hover to cruise mode and back at safe altitudes. Once in the cruise mode, these UAV types benefit from the same advantages as ordinary fixed-wing models. Figure 3 shows UAV types commonly used in the forestry sector.

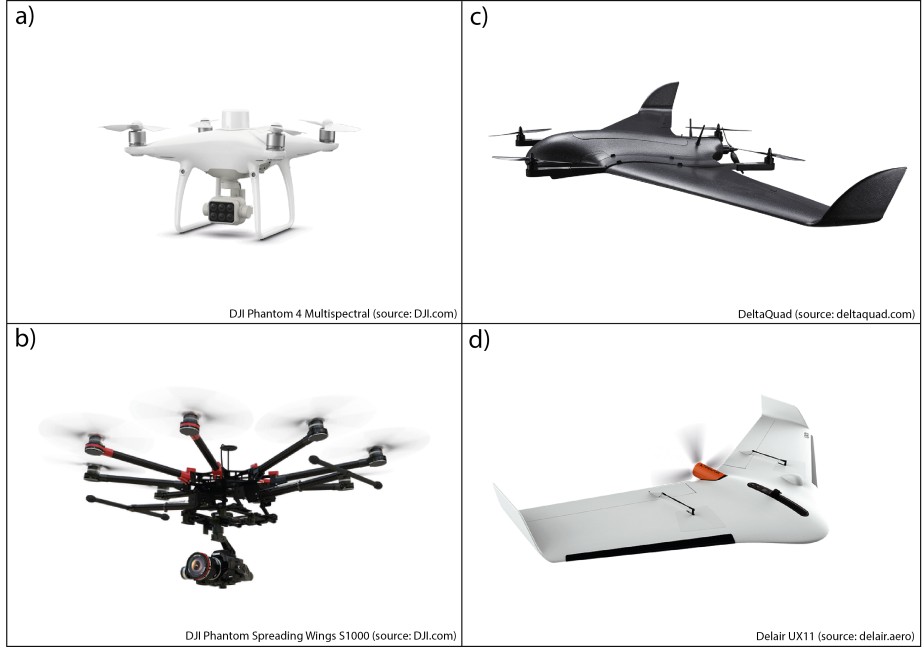

**Figure 3.** Different UAV types, commonly used for civilian applications: (**a**) rotary-wing (quadcopter), (**b**) rotary-wing (octocopter), (**c**) Hybrid VTOL solution, and (**d**) fixed-wing.

Technological innovations and developments have led to the increased use of UAV-RS. Contributing factors include (1) cheap, light, and miniaturized off-the-shelve UAVs and payloads [27], (2) the development of sophisticated image-matching algorithms such as semi-global matching (SGM) [39], which have paved the way for faster and more robust structure from motion (SfM)-based photogrammetry software [40], and (3) continued increases in computer capacity that made image matching feasible [41]. The emergence of SfM has enabled practitioners to produce their own georeferenced data products by using simple digital cameras [42]. In combination with UAVs, they become a powerful RS tool as a scale-appropriate application. In drone-based FHM, SfM photogrammetry provides

2D spectral data of tree crowns and corresponding 3D information that help to quantify biochemical leave compositions and structural attributes to draw conclusions about the tree's health status [21].

Typical payloads are mainly represented by optical sensors [43]. They range from consumer-grade RGB cameras to professional multispectral, hyperspectral, and thermal cameras, which all belong to the category of passive sensors. Less frequent but by no means less important, active sensors such as light detection and ranging (LiDAR) are also being applied. Figure 4 shows examples of typical sensors for UAV application for FHM.

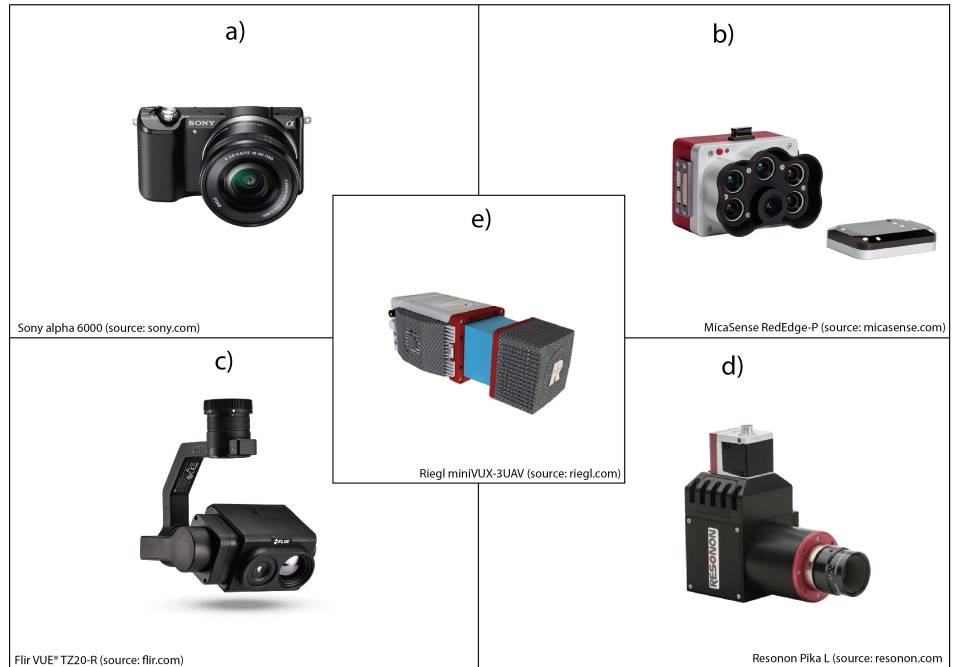

**Figure 4.** Different sensors for remote sensing applications that can be attached to UAVs: (**a**) RGB camera, (**b**) multispectral camera, (**c**) thermal camera, (**d**) hyperspectral camera, and (**e**) LiDAR sensor.

RGB cameras capture visible light approximately within the 400–700 nm band of the electromagnetic spectrum. Depending on the sensor and focal length, subdecimeter spatial resolutions can be achieved even from relatively high altitudes above ground when attached to drones [44]. It is common practice in RS to separate color channels to work with the individual bandwidths [45], for example, using different types of filters [46]. This allows the mathematical combination of color channels to derive band ratios that improve feature detection and provide foresters with valuable information [27]. Regarding vegetation analysis, additional bandwidths that reach beyond the visible light, such as near-infrared (NIR), are increasingly favored due to their higher sensitivity to plant pigmentation (e.g., chlorophyll and carotenoid content). Multispectral sensors consist of several separated bands (e.g., blue, green, red, red edge, NIR), usually covering the 400–1000 nm band of the electromagnetic spectrum. This allows the calculation of advanced vegetation indices that contribute to stress analysis and the evaluation of forest health [26]. Hyperspectral sensors can take the analysis of vegetation properties to a whole new level. They can cover similar spectra as multispectral sensors but significantly differ in band numbers and widths. Some consist of hundreds of narrow bands representing the electromagnetic spectrum in a much more detailed manner [47]. Thermal imaging sensors perceive emitted radiance typically between 7500 and 13,500 nm. Assuming an accurate calibration, pixel values can reliably be converted into temperature [25]. Due to technical limitations of optical sensors, the number of narrow bands of multispectral and hyperspectral cameras is coming at the expense of a lower spatial resolution [48,49]. To overcome this problem, pansharpening has become a widely used technique in RS with a variety of different pansharpening algorithms [48–50]. In the process, multispectral or hyperspectral imagery

is merged with higher spatial resolution panchromatic imagery while preserving the multispectral/hyperspectral sensors' spectral information and the geometrical detail of the panchromatic imagery [51–53]. Today, UAV-based imagery is also used to enhance the spatial resolution of satellite imagery [54,55]. Manufacturers that build specialized cameras for drones even started to supplement their multispectral cameras with panchromatic bands to enhance the spatial resolution using pansharpening techniques [56].

The group of active sensors includes, e.g., LiDAR systems. LiDAR detectors send active laser pulses in the NIR spectrum to scan the environment. The signals are backscattered by the objects (e.g., canopy and/or ground), and part of the transmitted energy returns to the sensor. The elapsed time between transmission and reception is recorded and combined with positional information, resulting in detailed point clouds containing intensity and elevation measurements [32]. Important forest health indicators such as tree crown density, pattern distribution, or structural changes over time can be derived from LiDAR-based point clouds and indicate defoliation [57–59] or changes in crown architecture [60]. Besides FHM, these sensors are used in various other forestry applications using drones. The 3D reconstruction of trees using RGB imagery [15] or LiDAR-based solutions [61] allow estimations of inventory parameters such as diameter measurements [62] and tree heights [63] and tree species [64], which in turn enable above-ground biomass calculations [65,66]. Furthermore, other structural information such as forest canopy gaps can be retrieved [67]. Instead, multispectral and hyperspectral sensors' broad spectral resolution is rather used for complex vegetation properties such as chlorophyll content [47] or tree species recognition [68,69], but they are also useful tools for applications similar to those mentioned for RGB cameras. Thermal sensors are often used for wildlife [70] and forest fire [71,72] detection.

### 1.3. The Importance of Forest Health

According to the United Nations (UN), more than 80% of all land mammals, birds, insects, and plants inhabit forests. It is estimated that about one-third of humanity directly depends on forests and their products [73]. Forests provide essential ecosystem services and ecological functions, e.g., to meet social needs such as recreation, education, and conservation, as a source of fuel and wood, for water purification, for erosion control, and climate regulation through carbon storage [23,74,75]. Besides anthropogenic pressures from timber harvesting, land use, and pollution, forests are also exposed to an array of abiotic (extreme temperatures, droughts, fire, storms, snow, and ice) and biotic (e.g., pests and diseases) stressors. Moreover, some of these stress agents are likely to interact and are further exacerbated by ongoing climate change [76]. Seidl et al. [77] concluded that warmer and drier climate leads to a favorable environment for fire, droughts, and insects, whereas warmer and wetter conditions foster storms and pathogens. As healthy forests are more robust and resilient to disturbances, it is critical to maintain and bolster their key functions [1,11].

The term forest health cannot be defined unambiguously, as it reflects values from social, economic and ecological considerations [78]. Lausch et al. [11] provide an overview of a range of definitions of forest health. Within the scope of this review, forest health is defined as the absence or presence of symptoms caused by biotic and/or abiotic stressors at the individual tree scale. Hence, the severity of tree damage is measured by quantitatively collected crown features attributable to biotic and/or abiotic stress agents. Regardless of the causative factor, trees may show different internal and/or external symptoms. They are caused by stress-induced physiological and morphological responses of trees as a process of acclimation to environmental changes in an attempt to compensate for the loss of productivity [79]. These phenomena can be observed spectrally and structurally at different levels. In the early stages of stress, indistinct (previsual) changes in pigmentation composition of leaves (e.g., chlorophyll and carotenoid) are caused by a change of activity of biochemical processes. For example, less activity in photosynthesis may occur [80]. These subtle symptoms are typically not visible to the human eye [81]. In advanced

stages, prolonged exposure to stress can cause irreversible damage, including discoloration, wilting, defoliation, and eventually lead to the tree's death (see, e.g., [82–84]).

### 1.4. Objectives, Limitations, and Review Structure

This review is intended to provide a comprehensive overview of the application of drones as an emerging tool for FHM over the past decade (2012–2021). It represents a selection and systematic analysis of the peer-reviewed literature related to the UAV-based detection of biotic and abiotic stressors, including tree mortality, and reveals early efforts towards UAV-based FHM. The goal is to uncover trends, research gaps, recent advances, technical challenges, and synergies to other applications or RS platforms. This review is intended to facilitate, for example, the design of experiments or acquisition strategies and the selection of appropriate measurement tools to detect specific stressors. Relevant literature is listed in a structured manner so that problems or shortcomings that have already been encountered can be addressed directly. Note that papers regarding the extensive topic of UAV-based forest fire detection were explicitly limited to the use of drones to assess forest health (damage and burn severity) following postfire events. This review is addressed to researchers, forest managers, and practitioners who are key drivers in advancing UAV-based forest health research and serves as a resource of ten years of relevant science to support their ambitions.

The organization in the remainder of this paper is as follows: Section 2 documents the creation of the literature dataset, its filtering, and categorization according to technical information and the causative agents of tree stress. In Section 3, results of the analyses are presented. In Section 4, the results are discussed, and technical issues addressed by the authors of the literature dataset are compiled. In Section 5, the main outcomes are finally summarized, and a conclusion is drawn.

## 2. Materials and Methods

### 2.1. Literature Search, Filtering, and Information Extraction

This subsection documents the dataset creation and the criteria the articles must meet to be included. The Web of Science (WoS) search engine was used to filter the appropriate literature to create a literature database. In order to narrow down the search, the criteria were chosen to be a combination of relevant terms or abbreviations describing the uncrewed carrier platforms, generic terms representing the field of forest science as well as keywords frequently being used in the specific application area of forest health RS. The search scope was further extended by the search operator "*" that can be attached to the keywords to retrieve other related words with variant zero and multiple characters (e.g., the keyword "disease*" includes diseased, diseases, etc.). Consequently, the combination of keywords covered ["uav", "uas", "rpas" OR "drone"] AND ["forest"] AND ["stress", "disease", "pest", "health", "defoliation", "dead", "disturbance", "infest", "infect", "insect", "pathogen", "damage", "drought", "abiotic", "decline", "severity" OR "mortality"]. The main search was limited to the period from January 2010 to December 2021 to see if there had been any earlier publications. However, the earliest publication found that met the selected criteria was from 2012. The search descriptors were entered in a common "Topic Search" to reduce the searchable fields to title, abstract, and keywords. Additionally, WoS-internal filters were activated to extract articles, proceedings papers, and book chapters in the English language only. The bibliographic dataset filtered as described was imported as a text file for further processing. Based on the unique WoS-ID, duplicates were removed using a Python script. By reading the titles and, where not explicitly clear, the abstracts, obviously off-topic papers were manually discarded. The remaining literature was downloaded and read in full. During the research, seven additional scientific papers were added that were not listed in the WoS directory or were not included in the keyword entry but were found to be relevant by the authors. A total of 99 research papers were finally considered relevant to the review's topic. The full list including all authors can be found in Appendix A. Data handling, statistics of extracted information and results

visualization was done via Python. A flowchart of the approach of the database search can be seen in Figure 5.

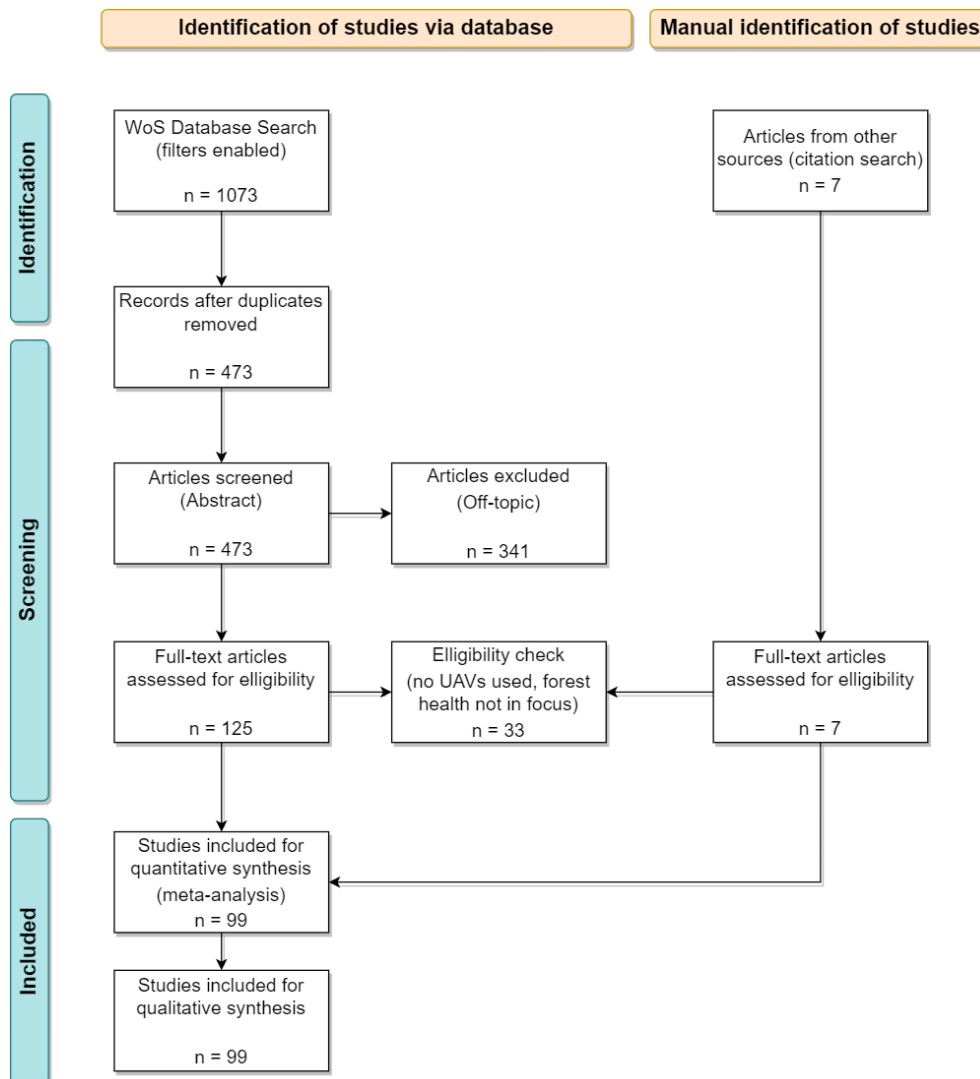

**Figure 5.** PRISMA flow diagram depicting the different phases of the bibliographic dataset creation (*n* = number of papers).

*2.2. Dataset Structuring*

This subsection describes the approach to structure the extracted information from the gathered literature. Once all papers had been read, an information extraction was performed in a two-step procedure. In the initial step, an overview table was created to list general information about authors, title, journal, publishing year, citation count, and a short summary of the content. The purpose of the table was to provide the authors of this review with an overview of the entire literature dataset and a first rudimentary summary of the content of the articles to ease a topic-specific assignment. Additionally, the overview table was used to sort the articles in descending order by year of publication to facilitate a systematic analysis and a more detailed information extraction. In a second and more extensive step, technical and strategic details on UAV and sensor type, flight parameters and revisit times, radiometric and geometric calibration techniques, data products, and data analysis methods were retrieved from the research papers. A particular focus was placed on information related to UAV-mounted sensors, complementary data from other sources, and machine learning algorithms for image detection and classification. Moreover, a comprehensive set of details on nontechnical information regarding forest health, tree species,

and stressors described in the papers were extracted. This information was then separated into two main subjects:

1. Technical and Strategic Information
   Technical details and strategic approaches common to all papers were subdivided into data acquisition, data processing, complementary data, and data analysis techniques with a focus on machine learning algorithms for image classification.
2. Investigated Tree Species and Biotic and Abiotic Stressors
   Information on investigated tree species and biotic and abiotic stress agents were summarized. Based on these details, the peer-reviewed papers were categorized into three main topics: (1) biotic stressors, (2) abiotic stressors, and (3) unknown/stress in general. Details on biotic stressors were further split into insects, diseases, and phytoparasites for clarity. Abiotic stressors were organized together due to the limited data. If the causal target stressors were unknown or not specified, they were grouped as unknown/stress in general. Papers in which the authors identified multiple stressors were listed in their respective categories or subcategories for completeness.

### 3. Results

This section describes the analysis of the previous information extraction from all collected works. Details regarding technical and strategic information on drone data collection, data processing, complementary data from other RS platforms, and data analysis techniques are documented. Biotic and abiotic stressors are then examined, and the tree species that are primarily affected are discussed.

Even though the literature search included publications from 2010, the first paper relevant to the subject of UAV-based FHM was published in 2012 [85]. A trend analysis of the peer-reviewed papers on UAV-based FHM can be seen in Figure 6. The most significant increase in articles occurred between 2017 and 2018, from 5 to 18 publications, respectively. Since then, the number of papers has been at a higher level with a continuing upward trend, peaking in 2021 with a total of 24 publications.

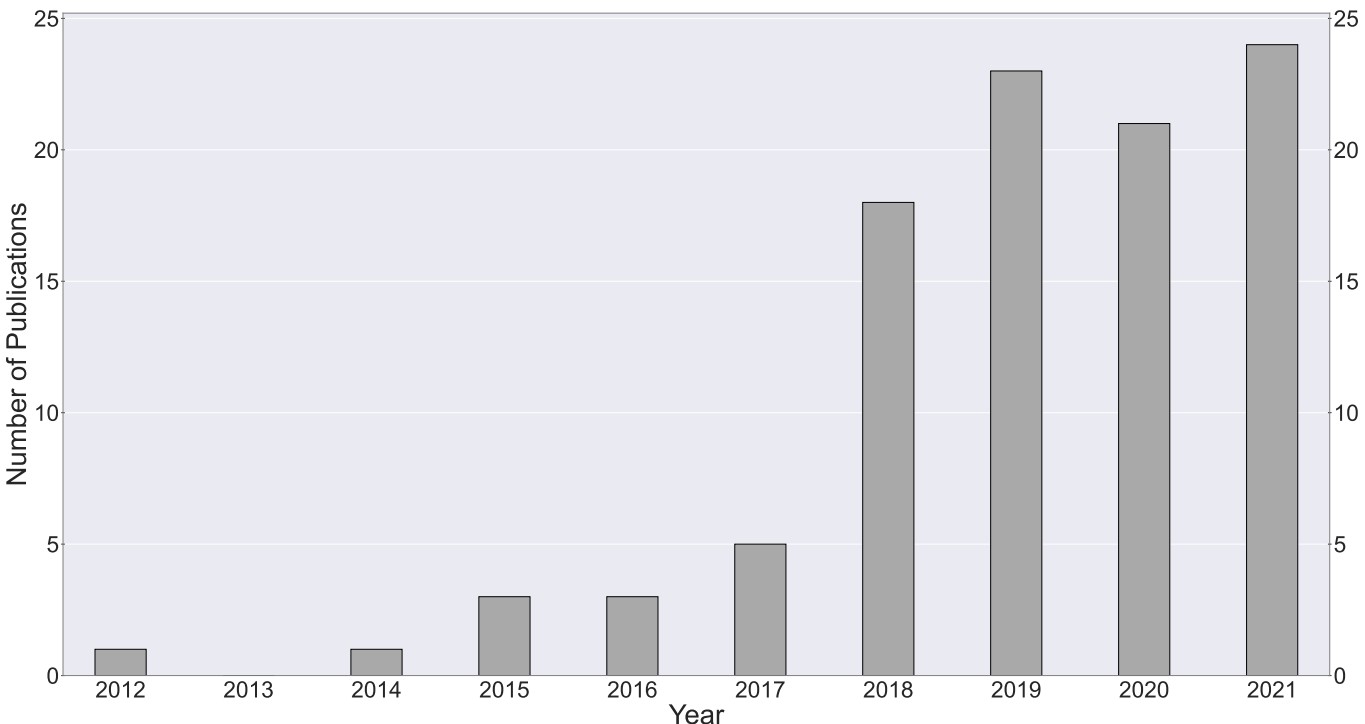

**Figure 6.** Bar graph of peer-reviewed papers related to UAV-based forest health monitoring published between 2012 and 2021.

*3.1. Technical and Strategic Information Extraction*

3.1.1. Data Acquisition: UAV Types, Sensors, Flight Planning, and Monitoring Strategies

Mainly multirotor systems were used as carrier platforms (77 papers). Some researchers (18 papers) used fixed-wing UAVs. In two studies, both multirotor and fixed-wing systems were applied, with the latter to achieve a more extensive area coverage [86,87]. A detailed comparison of the different platform characteristics was not considered. In one study, a single-rotor helicopter was applied [34].

A trend analysis of sensors attached to UAVs in the reviewed papers can be seen in Figure 7. In five studies, RGB cameras were modified with a filter to cover the NIR range of the electromagnetic spectrum [88–92]. For simplicity, such cameras were considered multispectral sensors. The more commonly used passive sensor combinations RGB and multispectral and RGB and hyperspectral were separately grouped. The latter also involved papers where the combination consisted of other additionally used cameras. All other passive sensor combinations were grouped into sensor combination (passive). Papers in which active and passive sensors were utilized were grouped into sensor combination (active and passive). The most commonly used sensors included RGB and multispectral cameras, which accounted for 90% of the whole dataset and were often used in combination or less frequently combined with other passive and active sensors. Along with the surge in publications, the number of sensors used of course also increased rapidly in 2018. However, the variety of sensors and their combination was clearly growing as well. Hyperspectral cameras have been applied more frequently since 2018, and LiDAR sensors since 2019. Nevertheless, RGB and multispectral cameras have dominated the use cases since 2017. The least represented sensors were thermal and gas sensors, with the latter only recently being used in a single study [93]. In about one-third of the studies, multiple sensors usage was described, and most often, the sensors were attached simultaneously to the same UAV. One study stood out, in which four cameras were mounted on the platform, including two oblique RGB, one multispectral (including thermal), and one hyperspectral camera [94]. In all but two studies [93,95], passive sensors were used.

Information about flight parameters, acquisition time, and weather conditions during the flight campaigns, if available, were also analyzed. It showed that, on average, relatively high image overlaps (80% front and 70% side overlap) were applied and a flight altitude of about 120–160 m was usually planned. The mean ground sampling distance (GSD) of RGB, multispectral, and hyperspectral sensors over the entire dataset was about 4 cm, 10 cm, and 25 cm, respectively. Weather conditions during data acquisition were mostly described as sunny, with low winds flown within one to two hours around the solar zenith. Some authors presented a calibration workflow under varying illumination conditions and pointed out problems during image processing with changing weather characteristics [92,94,96].

In 24 studies, multitemporal data with varying revisit frequencies were obtained. Repetitive UAV surveys of the same area were designed to collect time series as a database for change analysis. Hence, the scientists analyzed spectral and structural changes over time to assess mechanical crown damage [97,98], fire damage based on pre- and postfire data [99–101], phenological differences [89], and different stages of stress-induced symptoms evident in the tree canopy [92,94,102–112]. The primary period of FHM typically lay within the growing season. In some cases, data acquisition was carried out not only during the summer (leaf-on) but also in the winter (leaf-off) season to optimize the detection of the evergreen mistletoe (*Viscum album*) [113]. Others used the opportunity to get a clear view of the ground, enabling the creation of an improved photogrammetric digital terrain model (DTM) [114]. Only in six papers was the collection of UAV data over a period of more than a year described [97,98,105,106,108,115]. The longest and most comprehensive time series covered five years of monthly drone flights to study the influence of heavy rain events on tree fall and branch fall rates [97].

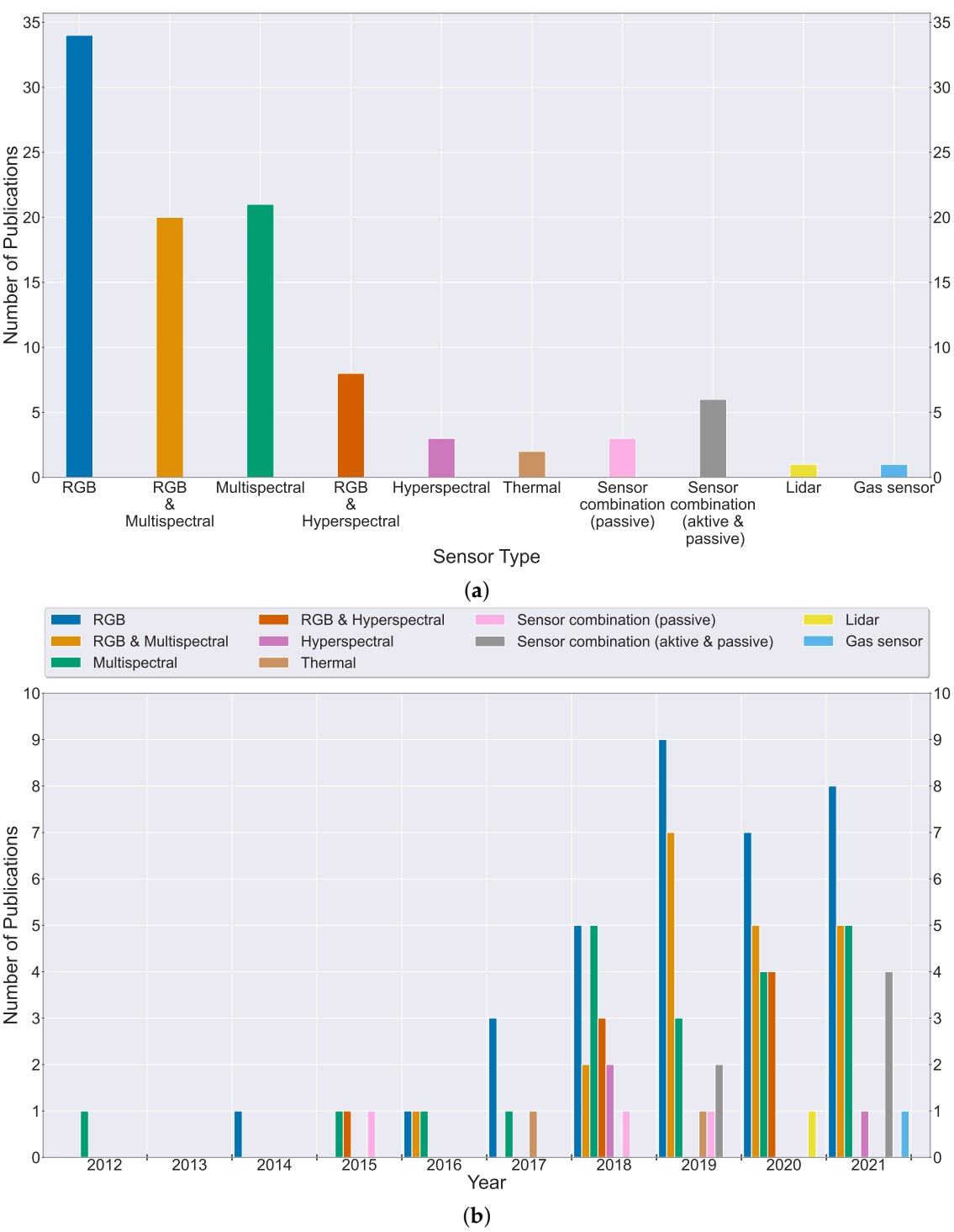

**Figure 7.** (**a**) Bar chart showing the distribution of different types of UAV-mounted sensors used for forest health monitoring in the literature dataset. (**b**) Bar chart showing the number of different types of UAV-mounted sensors used for forest health monitoring per year over the selected publication date range.

### 3.1.2. Data Processing: Data Products, Radiometric/Spectral Calibration, and Georeferencing

In most of the reviewed papers, researchers generated RS products from systematically acquired drone imagery and less frequently from UAV-based LiDAR metrics. The products included SfM-based or LiDAR-derived point clouds, which in turn served as the basis for digital surface models (DSMs), digital terrain models (DTMs), as well as canopy height models (CHMs), the latter representing tree heights by subtracting the DSM from the

DTM. The orthomosaic is by far the most widely used product when deploying imaging sensors, particularly in spectral analysis for assessing the health of trees. In some cases, spectral data were enriched with structural information derived from point clouds, DSMs or CHMs based on either LiDAR [116–118] or image data [96,102,119,120], further improving classification results. In a few papers, the raw drone images were directly analyzed without further processing [44,108,121–126]. To create photogrammetric products, the researchers predominantly implemented commercial SfM software such as Agisoft Metashape (Agisoft LLC, St. Petersburg, Russia) and Pix4D (Pix4D S.A., Lausanne, Switzerland). SfM is able to create a point cloud with true 3D information for the points from multiple overlapping images, similar to that one obtained from LiDAR. According to Iglhaut et al. [21], there are three critical stages of SfM:

1. Feature detection and matching of the corresponding key points in overlapping imagery;
2. Performing bundle adjustments to estimate intrinsic (e.g., focal length and lens distortion) and extrinsic (pose) parameters of the camera to compute a sparse 3D point cloud;
3. Multiview stereo matching to generate a dense point cloud of the scene. Through interpolation of the point cloud, rasterized spectral (orthomosaic) and structural data (DSM, DTM) can be generated for cell-wise statistical analysis.

To ensure radiometry uniformity across pixels and time, radiometric calibration needs to be performed prior to the SfM process [26]. This is recommended to compensate for changing illumination conditions and to generate consistent orthomosaics to maintain comparability [127,128]. In about 45% of the reviewed papers, a radiometric calibration process was described exclusively if multispectral, hyperspectral, or thermal cameras were used. The researchers mainly took reference images of single or multiple reflectance targets in the field, with known or directly measured reflectance values across the spectrum of light, typically using the empirical line method (ELM) for calibration during image postprocessing. Occasionally, the target information was enhanced with irradiance values taken from light sensors attached to the UAV [22,94,109,120,123,129–132], which were sometimes also used as the only calibration source [87,99,110,133]. In another study, the total incoming irradiance was simulated using data from a sun photometer [85]. Klouček et al. [92] implemented flat field corrections due to absent calibration measurements. Some researchers employed additional calibration strategies as illumination conditions changed during ongoing UAV flights. These included histogram matching [123] and radiometric block adjustment [34,94,134]. Moreover, the bidirectional reflectance distribution function (BRDF) was modeled to account for disturbances caused by the object reflectance anisotropy [134,135]. In contrast, thermal cameras were calibrated against a thermally controlled flat plate blackbody radiation source under laboratory conditions [136,137]. Intracrown illumination variability was treated with shadow removal techniques to reduce adverse effects on further spectral analysis. To separate shaded from sunlit pixels, histogram thresholding [138], index thresholding [129], and fractional selection of the brightest pixels within the crowns were performed [94,96,134,135]. Lin et al. [116] proposed an interesting approach to extract the shadow distribution of tree crowns in a three-dimensional manner based on a LiDAR point cloud and a 3D radiative transfer model.

During the SfM process, the data can be georeferenced based on positional information of the aircraft or reference targets. In fact, in more than 43% of the reviewed papers, the deployment of ground control points (GCPs) was described to georeference the UAV-based data products accurately. It is important to note that since 2019, there has been an emerging trend towards direct georeferencing using differential GNSS (D-GNSS) modules onboard UAVs. That enables a direct or postprocessed positional correction using real-time kinematic (RTK) or postprocessed kinematic (PPK) solutions. Two papers from 2019 [99,139], two from 2020 [94,140], and five from 2021 [100,112,130,141,142] substantiate this recent trend. Furthermore, other reference sources were used for geometric correction. Liu et al. [140], for example, used another UAV model equipped with a D-GNSS to create highly accurate RGB orthomosaics, which in turn were used for the coregistration of the hy-

perspectral orthomosaics taken with the other drone, while Otsu et al. [143] georeferenced their UAV data based on Landsat imagery.

### 3.1.3. Complementary Data: Fieldwork and Traditional Remote Sensing Platforms

In more than 75% of the reviewed papers, terrestrial field data collection, usually for ground truth purposes, was documented. Mainly topic-specific observations were described regarding the health status of trees. Notably, the presence of specific stressors and the severity of their impact were evaluated. The crown is most commonly the primary indicator to estimate tree condition. Visible symptoms such as discoloration, wilting, defoliation, and mechanical damage were registered, in some cases also the occurrence of lianas [144,145]. Other commonly surveyed data were basic inventory metrics (tree height, DBH, crown diameter and shape), tree species, tree locations, and GCPs. At the stem level, less pronounced symptoms confirmed the suspected stressors. They were also used to assess early stages (e.g., "green attack") of affected trees that cannot clearly be identified by observing the tree crown. For example, trees were examined for signs of early bark beetle infestations (resin flow, entrance holes, and boring dust) [90,120,130] and in doubtful instances also by removing the bark to expose potential egg galleries in the phloem [96].

Moreover, needle or leaf samples were taken to identify the presence of nematodes [106,142,146], to quantify pigments under laboratory conditions [85,105], and to assess their water content and spectral properties [123,136]. Soil sampling was conducted to determine possible origins of nutritional deficiencies [109] and to measure the field capacity [147]. Ground-based photographs were taken to calculate canopy cover [148], for the documentation of weather conditions [137], and for an improved categorization of pest infestation [44,118,142,146], disease [122], and fire [149] severity classes. Smigaj et al. [136] collected data from intratrunk water flow, canopy temperature, soil moisture, and incident and reflected light using an array of sensors. Other measurements were undertaken to derive ring widths [150], stomatal conductance [147], the presence of root rot [94], as well as gas exchange and fluorescence at the leaf level [105].

In 19 articles, data products from other RS platforms were incorporated to enrich the data pool. The researchers mostly relied on optical satellite data and aircraft-based LiDAR. Regarding satellite imagery, open-source data from Landsat and Sentinel were favored, but commercial providers such as QuickBird, SPOT6, and RapidEye were also used. For brevity, only a few articles were referenced, and the differences in synergies between UAV data and other RS sources were summarized. In some of the articles, the interpretation results of UAV-based data and from other RS platforms were solely used for the comparative analysis. Regarding satellites, Pádua et al. [151] investigated the complementary use of Sentinel-2 and drone imagery for postfire monitoring, while Dash et al. [104] incorporated RapidEye imagery to detect physiological plant stress. RS products from aircraft and drones were used to compare SfM-derived elevation information [152] and to evaluate their feasibility for FHM in vast boreal forests [96].

Beyond comparison, Viedma et al. [95] crossed Sentinel-2-based fire severity rates with UAV-based LiDAR point clouds to improve tree-level interpretation using structural information to additionally distinguish between canopy, midstory, and ground fires. In other studies, UAV-based data were used to calibrate models to exploit the added value of multiscale approaches further. Landsat data were calibrated with drone imagery to detect physiological plant stress [143] and to evaluate fire severity [149]. Campbell et al. [153] combined UAV-based RGB imagery with airborne laser scanning (ALS) data to calibrate Landsat imagery and scale up local estimates of tree mortality to larger areas. In one study, imagery from satellite and UAV were combined to densify a time series to monitor mangroves damaged during a winter freeze event [98]. Data fusion was also applied to normalize SfM-based DSMs using elevation models from ALS [89,134,137,152,154]. It is important to note that more elaborate data fusion techniques such as pansharpening were not found to be used in the literature dataset.

### 3.1.4. Data Analysis: Image Segmentation and Machine Learning Techniques for Image Classification

In about 44% of the reviewed articles, specific algorithms were implemented for automatic tree detection and image segmentation before classification tasks. The researchers utilized commercial software such as eCognition Developer (Trimble Geospatial, Westminster, CO, USA), MATLAB (MathWorks, Natick, MA, USA), and ENVI (Harris Geospatial Solutions, Boulder, CO, USA), as well as open-source solutions, namely R (using the lidR, itcSegment, rLiDAR and ForestTools packages), Orfeo toolbox, and Python (using the scikit-image package) for these tasks. Tree detection algorithms were mostly represented by local maxima filtering and less frequently by image binarization. Region growing and watershed segmentation were generally used for crown segmentation. The algorithms were applied to CHMs and point clouds based on LiDAR and SfM (or fused products), as well as orthophotos. Some papers provided a comparative analysis of different segmentation algorithms [22,95,155]. Minařík et al. [120] tested several segmentation techniques on different photogrammetric data products, while Lin et al. [116] compared the applicability of hyperspectral imagery and LiDAR point clouds for image segmentation.

Importantly, species recognition (17 papers) was sometimes part of the analysis or even the focus of the research [34,89]. Abdollahnejad and Panagiotidis [102], for example, tested classification results related to tree health with and without consideration of species differentiation.

In more than half of the articles (50), machine learning techniques were applied to classify individual tree health. In most of the other papers, a regression analysis was performed. In only three articles, researchers implemented physically based radiative transfer models [85,116,117]. In the remainder of this section, only machine learning practices for classification are discussed in detail. Nonparametric approaches represented the most used techniques with decision trees and the related classifier random forest (RF), as well as support vector machine (SVM), and k-nearest neighbors (k-NN). Instead, parametric approaches were represented by the maximum likelihood classifier (MLC) and logistic regression models. Interestingly, researchers have increasingly used deep learning architectures based on artificial neural networks (ANNs), mainly represented by convolutional neural networks (CNNs), since 2018, even more often than the popular RF classifier (see Figure 8). Other architectures related to ANNs were multilayer perceptrons [99] and autoencoders [144]. A comparison between different machine learning classifiers was also carried out [119,130,138,142,144,156,157]. The most comprehensive comparative analysis was performed by Wu et al. [111]. They tested nine traditional machine learning classifiers and four deep learning models against each other.

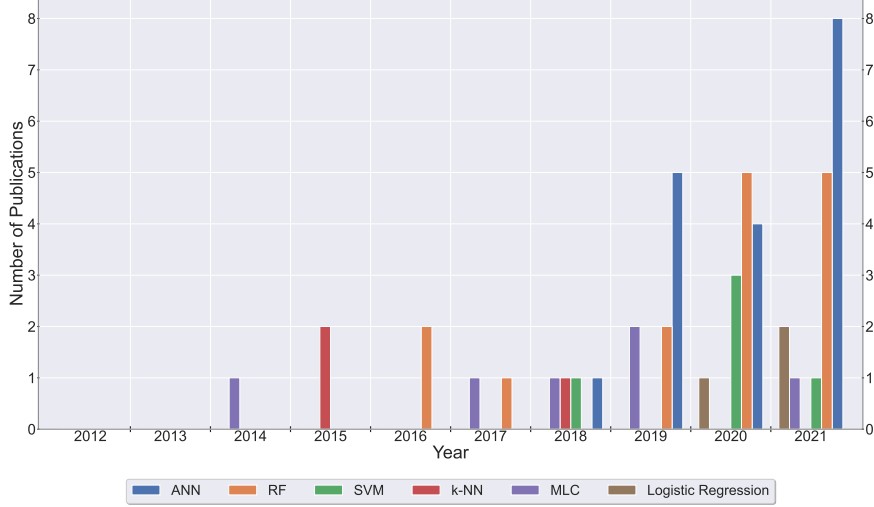

**Figure 8.** Distribution of different machine learning techniques used per year for image analysis over the selected publication period.

### 3.2. Tree Species and Stress Agents

The authors devoted most of their attention to conifers (70%), which were most often subject to biotic and abiotic stress factors. In particular, several pine species, Norway spruce *(Picea abies)*, and different fir species were investigated. Broad-leaved trees were particularly represented by oak [88,107,123,158] and eucalyptus [100,101,108,133,159] trees. Mangroves [98,139] and (sub)tropical [91,97,144,160,161] forest ecosystems were also examined. Biotic stressors clearly were the main concern with almost 70% documented throughout the entire dataset. Particularly insects (33.3%) and diseases (31.3%) were mainly investigated by the researchers. In comparison, abiotic agents accounted for 24.2% of stressors (Figure 9).

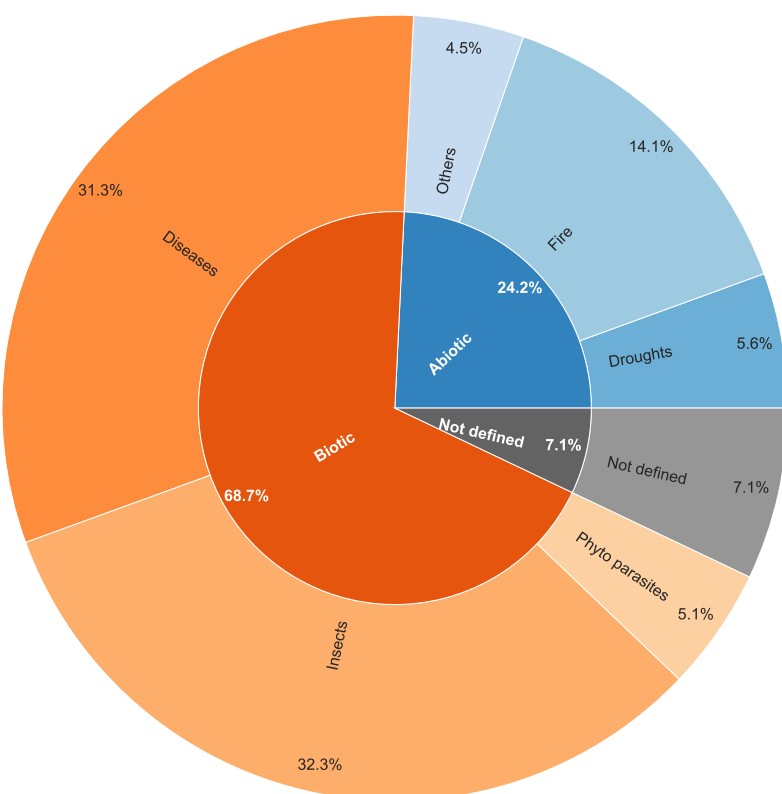

**Figure 9.** Pie chart of the distribution of different types of stressors addressed in the literature dataset.

#### 3.2.1. Biotic Stressors

(a) Pests

The insect species researchers dealt with belonged predominantly to the order of *Coleoptera* (72%), led by various bark beetle species *(Ips typographus* L., *Ips acuminatus, Ips sexdentatus, Pityogenes chalcographus, Polygraphus proximus Blandford, Dendroctonus Brevicomis and Dendroctonus mexicanus Hopkins)* followed by the pine shoot beetle *(Tomicus piniperda)*, the eucalyptus longhorned borer *(Phoracantha semipunctata)*, and the oak splendor beetle *(Agrilus biguttatus)*. The order of *Lepidoptera*, represented by the pine processionary moth *(Thaumetopoea pityocampa* and *T. wilkinsoni)* and the Chinese pine caterpillar *(Dendrolimus tabulaeformis)*, formed the other major threats by pests to coniferous forests. Other insect species belonged to the order of *Hemiptera* (Magicicada) and *Hymenoptera* (Oriental chestnut gall wasp).

Among the most relevant, the authors selected a few works, which are briefly described in the following. Näsi et al. [134] presented the earliest publication using UAV-based hyperspectral imagery to investigate bark beetle infestation *(Ips typographus)* at tree level. A machine learning algorithm was used to divide trees into three classes: healthy, infested, and dead. The best intraclass separability was reached using three different vegetation

indices. Klouček et al. [92] acquired multispectral time series to retrospectively detect early stage ("green attack") bark beetle infestations in a spruce forest. They yielded increasing overall accuracies (OA) of 78–96% with advancing tree decay over time. Instead, Cessna et al. [96] investigated the added benefit of SfM-based point clouds to put multispectral information into a structural context for the improved early detection of bark beetle attacks with comparable accuracies (OA = 78%, kappa = 0.64). Koontz et al. [22] conducted drone flights collecting RGB and multispectral imagery at 32 sites to link large-scale environmental gradients to local forest structure and composition to investigate tree mortality patterns. The research was initiated by an unprecedented mortality event during a prolonged drought in *Pinus ponderosa* populations. Previously weakened trees had been additionally infested by bark beetles *(Dendroctonus breviscomis)*, further intensifying mortality rates. The trees were classified as live/dead and host/nonhost with accuracies of 96.4% and 71.8%, respectively. It was found that greater tree density, proportion, the dimension of *P. ponderosa* trees, and, more importantly, a climatic water deficit (CWD) resulted in a higher tree mortality. Interestingly, they also demonstrated CWD/host-size interactions and concluded that larger trees were preferably attacked, especially in hotter and drier sites.

A very different approach was described by Schaeffer et al. [110]. They manually sampled bark-beetle-damaged trees in UAV-based orthomosaics to determine threshold rules for training classifiers based on cellular automata. A single study was devoted to the localization of bark beetle occurrences using a semiconductor gas sensor attached to a UAV to detect monoterpenes that could indicate an early stage of infestation [93]. Alpha pinene could be detected as an artificial source in the field. However, environmental factors and technical barriers compromised verification under natural conditions. Hence, a proof of principle above forest stands with trees in an early infestation stage has yet to be established. Zhang et al. [162] investigated defoliation caused by *Dendrolimus tabulaeformis Tsai* using hyperspectral imagery. They tested different dimensionality reduction algorithms and established a piecewise partial least-squares regression model reaching a high damage assessment accuracy at tree level of 95.23%. Lin et al. [117] proposed a three-dimensional radiative transfer model to assess damage levels caused by the pine shoot beetle (PSB, *Tomicus* spp.). Due to the vertical heterogeneity of forest trees, one-dimensional radiative transfer models are not sufficiently qualified. For this reason, they acquired UAV-based LiDAR data and in situ measurements to reconstruct three-dimensional damaged forest scenes. The reflectance of the forest scenes was then simulated by a radiative transfer model and validated using both Sentinel-2 and UAV-based hyperspectral imagery with overall good performances. They concluded that 3D models had great potential to accurately estimate PSB damage for forests. All papers regarding pests have been collected in Appendix A Table A1.

(b) Diseases

Here, researchers dealt with forest diseases caused by a range of pathogens. In half of the articles (16 papers), the pine wilt disease (PWD) caused by nematodes was the focus of the research. Fungi represented the second major source of tree infection. In this context, the researchers investigated the fungal pathogens *Dothistroma septosporum* [136,137], *Armillaria ostoyae* [90], *Chrysomyxa rhododendri* [122], *Austropuccinia psidii* [163], *Raffaelea quercus-mongolicae* [107], *Hymenoscyphus pseudoalbidus* [164], and *Heterobasidion annosum* [94]. Other investigated pathogen effects were caused by bacteria (*Ralstonia* sp.) [159] and oomycetes represented by *Phytophthora cinnamomi* [109,115] and *Phytophthora alni* [89,119]. Acute oak decline was also reported [123,158], which is likely to be caused by an interactive bacterial pathobiome [165].

Regarding the most interesting articles, Dash et al. [103] simulated a disease outbreak by treating selected trees of a pine plantation with herbicide. The area was monitored several weeks before and after the treatment using a multispectral camera attached to a UAV. Simultaneously, ground surveys were conducted. The timing of the pine trees' first stress response that was detected based on discolored needles correlated with the field assessments. The normalized difference vegetation index (NDVI) proved to be the most

sensitive vegetation index to physiological changes in leaf pigments. However, the best results (k = 0.694) were obtained using a combination of a three-variable model (including NDVI) implemented in an RF classifier. Building on the first study, the authors compared the same drone data with RapidEye satellite imagery [104]. They found that the NDVI was the most sensitive to discoloration in their UAV data. In contrast, in the RapidEye imagery the normalized difference red edge (NDRE) was more sensitive for the same purpose. The study highlighted the utility of vegetation indices such as NDVI in combination with RF classifiers to detect tree stress for both UAV as well as RapidEye satellite data. Overall, the UAV data were more sensitive ($R^2$ = 0.84) to fine-scale instances of physiological stress than the satellite data ($R^2$ = 0.73).

Ganthaler et al. [122] investigated the needle bladder rust disease of the Norway spruce caused by the fungus *Chrysomyxa rhododendri*. They obtained oblique and nadir aerial RGB images of tree canopies using a drone. They automatically detected diseased and healthy needles by thresholding saturation, brightness, and hue and identified strong correlations between manually counted needle samples ($R^2$ = 0.88–0.98). Yu et al. [118] used UAV-based hyperspectral imagery and LiDAR data for the early detection of the pine wilt disease. Utilizing an RF classifier, they found the highest accuracy by fusing both sensors to distinguish five stages of infection severity (OA: 73.96%, kappa: 0.66). Xia et al. [44] instead used unprocessed UAV-based RGB imagery and showed that deep learning approaches represented by several CNN models were superior to the more traditional RF classifier for detecting the pine wilt disease. All papers on diseases have been collected in Appendix A Table A2.

(c) Phytoparasites

Other causal agents for biotic stress found in the literature dataset were the hemiparasitic mistletoe species *Amyema miquelli* [108] and *Viscum album* [113], as well as lianas [144,145,166], the latter sometimes referred to as structural parasites competing for light and below-ground resources [167,168]. Maes et al. [108] measured the tree canopy temperature of mistletoe-infected and noninfected eucalyptus trees *(Eucalyptus fibrosa and Eucalyptus moluc-cana)* as well as the box mistletoe *(Amyema miquelii)* foliage temperature, using a thermal camera attached to their drone. They found that mistletoe canopy temperature was up to two degrees lower than the infected host tree canopy temperature, which they concluded may be caused by higher transpiration rates of mistletoe. When incoming radiation peaked, these temperature differences were most pronounced. In the same conditions, they also reported that infested trees had higher canopy temperatures than noninfested trees, suggesting lower transpiration rates of host trees caused by the hemiparasite. Miraki et al. [113] focused on the automatic detection of host trees with mistletoe *(Viscum album)* infection, mostly in Persian ironwood *(Parrotia persica)*, using an RGB camera. As this mistletoe is an evergreen species, the authors took advantage of this characteristic and collected UAV data in foliated and defoliated tree conditions. They successfully detected infected and noninfected trees applying an RF classifier with various vegetation indices within the visible spectrum. The best results were reached by combining both leaf-off and leaf-on datasets with 87% and 76% overall accuracy using manually and automatically delineated tree crowns, respectively. All papers related to phytoparasites have been collected in Appendix A Table A3.

### 3.2.2. Abiotic

In the related literature of abiotic stressors, the researchers were primarily concerned with forest fires, particularly the assessment of postfire damage and tree recovery, as well as drought-induced tree stress. Other studies dealt with extreme temperatures [98], inundation stress caused by tidal activity [139], and the physical impairment of trees due to precipitation of snow [169] or heavy rainfall [97]. Among the most important papers, we chose the works of Hernández-Clemente et al. [85], D'Odorico et al. [105], and Campbell et al [153]. To the best of our knowledge, the earliest publication addressing physiological variations of pine trees with drones applicable for FHM dates from 2012. Hernández-Clemente et al. [85]

validated a physically based radiative transfer model for complex structures parameterized with detailed field measurements using drone-based multispectral imagery. They found that traditional vegetation indices were strongly influenced by structural variations when compared at the leaf and canopy level. Furthermore, they introduced a new vegetation index ($R_{530\text{ nm}}/R_{570\text{ nm}}$) that was significantly related to the carotenoid content at both leaf $R^2 > 0.72$; $p < 0.001$) and crown levels ($R^2 > 0.71$; $p < 0.001$), and which proved to be the most robust against structural effects compared to other indices tested. The high-resolution images enabled varying carotenoid content to be mapped at the tree scale to draw conclusions about the physiological state of pine trees exposed to arid conditions.

In the work of D'Odorico et al. [105], another vegetation index sensitive to carotenoid content was investigated using a modified multispectral camera attached to a UAV. They examined a long-term precipitation manipulation experiment established in 2003 by irrigating selected plots in a mature *Pinus sylvestris* forest. After 11 years, irrigation was stopped for approximately one-third of the irrigated areas. This resulted in a comparison of irrigated, irrigation-stopped, and untreated (control) plots. From 2019, 17 years after the area was set up, they collected drone-based multispectral data in three flights per year (spring, summer, and autumn) over two consecutive growing seasons. With this setup, the authors detected drought-induced tree stress and their post-drought recovery based on the photochemical reflectance index (PRI; Gamon et al. [170]). The control plots measured up to ten times lower PRI values, indicating higher stress levels than the irrigated plots. Moreover, they found positive legacy effects on sites where they had stopped irrigation five years before the first drone flights launched compared to the control sites which were never irrigated.

Campbell et al. [153] introduced a multisensor (RGB, LiDAR, and multispectral) and multiscale (UAV, crewed aircraft, and satellite) monitoring approach for mapping tree mortality in a piñon–juniper woodland in a model calibration framework. Based on historical evidence, the authors considered drought the most likely catalyst for the tree die-off. They combined high-resolution UAV-based RGB imagery for the interpretation of tree mortality at individual tree-scale (live, partially dead, and dead trees), airborne LiDAR data for individual crown segmentation, and determination of relative canopy mortality rates, as well as Landsat imagery to allow the extrapolation of mortality estimates to a broader spatial scale. All papers regarding abiotic stressors have been collected in Appendix A Table A4.

### 3.2.3. Stress in General or Unknown Cause

In a few papers, the authors did not specify the causative agent and investigated tree vitality in general. For instance, Khokthong et al. [148] assessed drone-derived canopy cover and evaluated its correlation with the mortality rates of palm trees, while Campos-Vargas et al. [156] tested a range of traditional and advanced machine learning algorithms to separate live from dead trees as well as forest gaps. Although focusing on biodiversity, Saarinen et al. [34] also explored the adequacy of UAV-based hyperspectral imagery to evaluate the health status of trees. They found that a vegetation index using a red-edge band of 742 nm and an NIR band of 900 nm was particularly appropriate for this task. All papers in which the cause of tree stress was not defined have been collected in Appendix A Table A5.

## 4. Discussion

In this section, the results are discussed in detail and technical advances but also research gaps and trends are addressed following the order presented in Section 3.

The trend results regarding the increase of UAV-based FHM related literature trails approximately one to two years behind the exponential trend of UAV-based forestry applications in general [171], which might explain the underrepresentation of FHM compared to other disciplines in the related literature [16,172]. However, this also shows that this topic is establishing as an individual application with rapid growth to be expected in the following years.

### 4.1. Data Acquisition

Regarding data acquisition, technological advances in hardware and software enable standardized approaches for flight planning and automated drone operations. This ensures that preplanned flight parameters are met to fulfill the requirements according to resolution, area coverage, lighting conditions, and possible SfM workflows [173], assuming photogrammetric products are desired. For other applications, a systematic and precise aerial coverage is most often requested. Due to their maneuverability and comparative ease of operation, and the ability of VTOL, most researchers turned to multirotor UAV models. Since most study areas were manageable in terms of size and only representative forest stands were selected, maneuverability and flexible operation seem more critical than extensive area coverage. In contrast, fixed-wing aircraft were used if an extended area coverage was required. The less frequent use of fixed-wing models could be due to the higher costs, higher planning effort, and complicated deployment. Surprisingly, hybrid VTOL fixed-wing models were used in none of the studies. These would significantly mitigate the disadvantages of fixed-wing UAVs, especially during take-off and landing, and do not require large open spaces. Once the UAV is manually maneuvered above the tree tops in the multirotor mode, all the advantages of fixed-wing aircraft come into play by transitioning into the cruising mode at safe heights. Considerably higher costs probably contribute to the fact that this type of UAV model does not yet serve a broader range of customers. Nevertheless, we believe that these models will play a more prominent role, especially in the forestry sector, to cover larger scale areas.

In most of the articles, the use of RGB cameras or their combination with other sensors was reported. When comparing different imaging sensors, the dominant presence of RGB cameras was not surprising. These are often the first choice due to their availability, customizability, high resolution, and low costs, providing good SfM results. The relatively high proportion of multispectral sensors compared to other forestry applications [172] also shows that the scientific community is particularly interested in detailed spectral analyses when it comes to FHM. Additional bands in the near-infrared spectrum allow for improved vegetation analyses. The growing share of hyperspectral cameras, despite their higher costs and processing complexity, underlines this assumption. Indeed, broad RGB and multispectral bands may inevitably average fine spectral components, which could still be discriminated by the multitude of narrow bands provided by hyperspectral sensors [47]. Increasingly affordable and miniaturized of-the-shelf multispectral cameras are supposedly a reason for the rapid spread. Multispectral camera and SfM software providers even work in collaboration on common workflows to simplify data import, and subsequent workload, including spectral calibration [174]. This trend facilitates an entry into UAV-based spectral imaging and will further boost the adoption of highly specialized cameras for FHM. Hyperspectral camera providers will likely emulate this strategy in the future. Multispectral cameras designed specifically for the forestry sector, representing only the most important bands and bandwidths, would also be conceivable. Most multispectral sensors currently available have been developed primarily for the agricultural sector.

Besides spectral details, structural information was primarily derived from photogrammetric products, but the use of more sophisticated LiDAR sensors is increasing. Quantifying spectral and structural characteristics of trees with UAV-RS allows the discrimination between different health classes of trees or tree species if differences in certain characteristics (e.g., biochemical, physiological, or geometric features) can be detected [12]. Drone data of a single date depict only the momentary condition of trees and thus allow only for limited conclusions to be drawn. Therefore, conducting aerial surveys with high-frequency revisit times and preferably over a longer time horizon is a significant advantage of UAVs. When spectral and structural RS information reaches its limits in distinguishing between healthy and dead trees or even different stages of diseased trees, seasonal changes in tree characteristics such as photosynthetic activity, physiological and geometric attributes, and changes in biomass could help [11]. Health progressions and deterioration or recovery can only be analyzed reliably with time series based on repeated data acquisition. Although the

high temporal resolution possible with drones is emphasized, there is only limited literature available that make advantage of this feature. In just 24 papers were multitemporal data acquired, and only 6% of data were collected over a period longer than one year. This obviously underexploited potential should be addressed more intensively in future works related to FHM, where research on plant dynamics is highly relevant. Particularly given the structural complexity of trees, long-term monitoring and high-frequency revisit times could reveal valuable additional information and potentially overcome technical and analytical limitations.

The prerequisite enabling photogrammetric processing using SfM software is a sufficient image overlap. Due to the complex structure of forests, a relatively high image overlap is recommended [15,21,173] and was adopted by the majority of the reviewed papers. Appropriate flight configurations can optimize the produced data volume. It is crucial to balance the amount of imagery but still achieve efficient image overlap to avoid drawbacks due to long processing times but simultaneously not compromise data quality [175]. The researchers barely addressed the issue of excessive data storage within the literature dataset. Only in one paper was redundant drone imagery filtered in real time to reduce processing efforts [124]. The same applies to the flight altitude, which is rarely experimented with. Seifert et al. [15], for example, showed that low flight altitudes could be beneficial for SfM-based forest reconstructions, especially in combination with high image overlaps. Whereas in Young et al. [176], this effect was not observed, and higher flight altitudes even led to improved tree maps. Recommended flight times and weather conditions were also reflected. Even though overcast skies allow for a uniform illumination due to diffuse lighting, the lowest possible atmospheric influences are conducive to spectral studies [21]. Therefore, clear skies were preferred throughout the literature dataset. Even though some cameras are bundled with on-board irradiance sensors, changing illumination during the data acquisition still poses a great challenge to calibration algorithms. Hence, fluctuating light conditions can have a major impact on data quality and therefore need to be studied more intensively. In general, flight parameters should be given more attention in future research as they have a direct influence on data quality.

Drone flights are subject to legal requirements, and the regulatory framework provided by each country has a major impact on data acquisition and flight parameterization. For example, European Union member states' maximum allowed flight height is limited to 120 m above ground level. Moreover, the UAV needs to be within the visual line of sight (VLOS) [177]. Thus, the coverage of larger areas strictly depends on local rules. Flights above the maximum allowed height and beyond the visual line of sight (BVLOS) usually require exemptions, which the respective air safety authorities must approve. UAV forestry applications may also be affected by flight restrictions over national parks and conservation areas or in the vicinity of energy and traffic infrastructure and might, on top of that, be subject to privacy laws [178]. These rules are essential to guarantee safe and legal drone operations while protecting the public interest. Since UAVs used in a civil context are relatively new, the respective legislation frameworks might be changing dynamically. The legal framework for drone operation should be further developed considering the operational use of drones, safety issues, and the protection of public and environmental interests, which requires thorough communication between stakeholders and policymakers.

### 4.2. Data Processing

Programmable processing workflows implemented in photogrammetry software allow for a fully automated SfM data creation. From the raw images to the final products, the entire process chain can be fully controlled, automated, and adapted to individual requirements. However, almost all reviewed studies heavily relied on commercial SfM software to create data products. This could pose a financial barrier and, considering the political situation, also a legal barrier to stakeholders, and it might impede research and technology transfer. For an overview of open-source SfM tools and algorithms, we refer to [179,180].

Spectral information is of pivotal importance in FHM and probably the most significant indicator for forest health in RS data [11]. Hence, orthomosaics from multiple spectral layers are the focus of research in this application field. Most studies produced orthophotos that were georeferenced and homogeneous, but much information that can be derived from perspective imagery (e.g., lateral tree views) are eliminated during the photogrammetric process. Thus, if one does not need a point cloud or orthophotos because one already knows the structure, not using SfM can also have great advantages as in some of the papers demonstrated. Too few of the studies focused on the raw imagery, which also have great potential for detecting tree damage.

Geometric tree data are used mainly for preliminary tree detection and crown segmentation and sometimes to improve interpretation results. The authors of the analyzed literature predominantly relied on SfM-based and less commonly on LiDAR-based point clouds. Photogrammetric point clouds are limited to the canopy surface in dense forests representing the visible surface with even higher point densities and details than LiDAR [21]. Problems in processing the data are mainly due to the high structural complexity of the tree crowns, which is often accompanied by wind-induced motions of leaves and branches. Especially with an insufficient image overlap, this can lead to image-matching problems. LiDAR-based point clouds often have the advantage of enabling the reconstruction of internal tree structures and the ground surface [181]. When directly comparing LiDAR and SfM-based point clouds to derive forest inventory metrics, LiDAR is often superior to detect subcanopy information [182]. However, this statement is only valid if the LiDAR system has a sufficient resolution (points per m²), able to represent the vertical structure of the tree as well as the ground. Nevertheless, SfM provides a cost-efficient alternative, making it a competitive and appropriate tool if laser technology is unavailable [21,183,184]. It would be conceivable to have one-time or infrequent LiDAR drone surveys kept up to date by regular SfM-based surveys [181]. Some of the authors pointed out that 3D geometric information can lead to better classification accuracies and improved interpretations of tree health. As stressors may act on different spatiotemporal scales, symptoms may first occur at the crown's lower or inner parts [155], making them hard to detect in images from the bird's view. The increasing use of LiDAR sensors in this application field suggests a trend in an appropriate direction that uncovers detailed 3D structures of trees. LiDAR should be more extensively incorporated into future research, especially with imaging sensors.

A critical step of the SfM processing chain is the radiometric and spectral calibration of the images. Radiant energy is received by the senor's pixels and saved as a digital number (DN). Usually, the goal of radiometric and spectral calibration is to convert DNs into reflectance values. In some of the reviewed papers, researchers pointed out problems with calibration techniques during overcast skies or varying weather conditions. This reveals that calibration is still a key issue and thus diminishes the opportunity to fly below the cloud cover, if a detailed spectral analysis is desired [101,185]. Recent developments and efforts by providers of photogrammetry software (e.g., Agisoft and Pix4D) already enable the integration of reference reflectance targets, and irradiance measurements show that this problem has been recognized and is being worked on. However, there is no consensus on standardized calibration workflows for fluctuating lighting conditions to create consistent reflectance maps [186]. We refer to Aasen et al. [26] for more information on this topic, including state-of-the-art radiometric and spectral calibration techniques. The introduction of multispectral active sensors, such as multispectral LiDAR for UAVs, might be a game-changer in this regard in the future, since it overcomes both the problem of the geometric reconstruction as well as the spectral calibrations due to inconsistent lighting conditions [187,188].

The potential of direct georeferencing concerning UAV-RS was already pointed out at an early stage of UAV development [189]. Today, with RTK/PPK-ready UAVs, an essential step towards operational UAV-RS has been achieved [190,191]. Distributing and recording GCPs is time-consuming, costly, and usually not feasible in hard-to-access forestry areas. In dense forests, the accuracy of handheld differential GNSS (D-GNSS) devices also suffers

due to signal interference from high vegetation [192]. On top of that, the GCPs are often poorly detectable in aerial imagery due to a lack of sufficient open space. The association of GCPs during photogrammetric processing can also only be automated to a limited extent and requires user interaction which further impairs the automation of the process chain. The distribution of a few GCPs for reasons of validation is recommended.

An important argument for using UAVs for FHM is that drone operators are also data providers and have complete control over the data processing workflows. This is a viable advantage over traditional RS platforms as there is no dependency on third-party data providers. In fact, this could bolster an open-source mindset and promote an open-data policy framework for the UAV-RS community.

### 4.3. Complementary Data Use

The high proportion of recorded fieldwork indicates how much importance is attributed to ground truth information for FHM. In many cases, field data were used for result validation. External symptoms detected from aerial imagery only cannot confirm most stressors. Field surveys will continue to play a central role in better understanding the complexity and interactions among multiple stressors on forest health. UAVs are instead seen as a complementary tool. A monitoring system that combines in situ terrestrial information based on expert knowledge and RS data should be aspired to [1]. Drone surveys can help to simplify and sometimes even substitute for specific tasks in the field, but a general replacement of ground surveys for FHM by drones is still far from conceivable. RS cannot sense specific forest health indicators, such as underground measurements, that necessitate terrestrial efforts.

According to Emilien et al. [193], the complementary use of satellite and UAV data can be divided into four categories with increasing synergies, respectively: (1) data comparison, (2) multiscale explanation, (3) model calibration, and (4) data fusion. Especially the latter two strategies, which are considered to have strong synergies, have been barely used in the reviewed articles and should be explored more closely to exploit the additional benefits for FHM. Sophisticated data fusion techniques such as pansharpening to merge complementary satellite and UAV images are generally underrepresented in UAV-related research [193] and were not sufficiently exploited within the literature dataset. Entire satellite archives such as Landsat data were made publicly accessible [194] and thus serve as a potential complementary data pool. Other Earth observation programs, such as the Sentinel space missions of the European Space Agency (ESA), have been designed from the outset with an open-data policy in mind. Combined with freely available cutting-edge processing tools for RS imagery [195], this opens new possibilities for forest-health-related science. These principles should be endeavored by the UAV-RS community. Future efforts should utilize products from different RS platforms to scale drone data to larger areas and improve interpretations, as some authors demonstrate. Standardization of UAV data and data structures fosters interoperability and should be pursued to facilitate data fusion to develop robust multisource and multisensor RS systems.

### 4.4. Data Analysis

Approximately 20 years ago, the ever-increasing spatial resolution provided by modern spaceborne imaging sensors had led to the questioning of purely pixel-based approaches for geospatial image analysis [196]. This has prompted a demand for research beyond pixels. Image segmentation is a crucial step of object-based image analysis (OBIA) and decisively determines the quality of the subsequent feature extraction and classification [197]. Using segmentation algorithms, researchers attempt to group pixels into meaningful image objects based on homogeneity criteria [198]. This was also reflected in our literature dataset. To a large extent, the researchers implemented automatic tree detection and crown segmentation algorithms. Typically, these algorithms were used on CHMs or the original point cloud. Minařík et al. [120] indicated that the point density derived from SfM was more critical for the segmentation success than the algorithm used and should be well

above 10 to 20 points per square meter. More complex forest structures with intertwined tree crowns may require LiDAR-based point clouds [199].

Automatic processes are essential to handle large amounts of data efficiently. The same applies to classification tasks. Machine learning has shown great potential for automated image recognition and classifying healthy and decaying trees. Many of the authors used RF classifiers either pixel-wise or on preliminary segmentation outcomes for the OBIA classification as it provides robust results. An emerging trend shows that RF is gradually being superseded by modern deep learning architectures such as CNNs. The main advantage of these techniques are the very high classification accuracies and the fact that those algorithms do not need any subjective and laborious feature engineering. However, the main drawback of CNNs is the enormous amount of sample data and processing resources necessary. Moreover, the complexity of deep neural networks reduces the interpretability of the input features to the degree that they become meaningless abstractions that can hardly be linked to real-world features [200]. In contrast, conventional machine learning classifiers such as RF operate resources efficiently [201], requiring only a fraction of sample data. Complex machine learning architectures and workflows hinder widespread adoption beyond the scientific community, requiring a more intensive knowledge transfer. Simple, robust, and competitive approaches for high-resolution drone imagery are urgently demanded [119].

### 4.5. Tree Species and Stressors

As expected, mainly conifers were researched, which is in line with the findings of [16]. The conifers were mostly impaired by biotic and abiotic stressors. They often represent fast-growing tree species and economically important assets that have been used for decades in plantations to optimize yields. Due to a reduced branch growth in dense monocultures, easy-to-process timber for large lumber mills can be produced for the mass market. High host densities also favor mass insect reproduction and the spread of diseases [22,202].

Therefore, it is not surprising that pests and diseases account for almost two-thirds of the literature dataset. Some authors addressed important cross-correlations of multiple stressors that may exacerbate the detrimental effects on forest health, and that are yet not sufficiently understood. With an advancing climate change, alterations of disturbance frequencies and intensities are to be expected in the future [203,204]. Early stress detection may lead to a better understanding and monitoring of forest health and improve prevention measures by identifying appropriate forest management strategies. Late physiological effects result in progressive changes in pigmentation and, finally, defoliation, which often represents irreversible damage to the status of trees [105]. Future research should intensify investigations on previsual symptoms at early stages of stress to evaluate tree recovery opportunities and develop strategies to prevent further decline. Moreover, efforts should be made to integrate essential prerequisites such as crown segmentation and species identification into the work process and include them in the evaluation of forest health. For this to succeed, it will be necessary to optimize the spectral and temporal resolution possible with UAVs and apply additional active sensors to acquire state-of-the-art 3D information.

### 5. Conclusions

Drones for FHM are getting increasingly popular as they have proven to serve as a cost-efficient and reliable RS tool for stress detection at the tree scale. This is also reflected in an ever-growing number of related literature works. In this review, we compiled and analyzed a comprehensive dataset of 99 papers regarding UAV-based FHM published over the last decade. To limit the scope of the research, we focused on publications dedicated to tree stress induced by biotic and abiotic agents. The increasing number of articles show that UAV-based FHM is in the process of becoming a mature forestry application field.

UAVs, associated technologies, and sensors are becoming cheaper, easier to handle, and more advanced. Technological advances are rapidly adopted and pave the way for an operational UAV-supported forest monitoring system and boost further adoption among

stakeholders. Onboard D-GNSS modules, for example, can drastically reduce or even replace the time-consuming distribution of GCPs. The number of multispectral cameras used in the field is relatively high compared to other forestry disciplines. This indicates the broader acceptance of specialized sensors and reveals that a greater technical effort is being expended for the sake of analytical precision. SfM processing has been established as a solid approach and delivers reliable data products providing insights into spectral and structural tree characteristics. Drone monitoring of tree vigor and plant dynamics has shown to be effective and robust, as the authors vividly demonstrated in their research. Nevertheless, we also identified a few drawbacks and research gaps that need to be overcome in further study:

(1) The use of hyperspectral cameras and LiDAR sensors must increase to exploit modern sensors' spectral and structural potential. Both sensor types are underrepresented but provide the most promising added value for FHM, in particular if used in combination. Optimized multispectral sensors for FHM need to be developed, which offer the most informative spectral bands tailored to the requirements of the analysis of forests. The miniaturization of multispectral LiDAR must be advanced for UAV applications.

(2) The flexible use of drones enables recordings with short revisit intervals and thus provides a high possible temporal resolution. Despite these advantageous attributes of UAVs, multitemporal and long-term monitoring is not sufficiently performed yet. The dynamic nature of forest ecosystems requires timely and repeated data collection to capture forest health degradation and recovery.

(3) A greater emphasis should be placed on early stress detection (previsual stage) to improve forest management strategies and to be able to respond to impending damage promptly.

(4) Currently, a fast advance of operationalization of UAV-based FHM is mainly hampered by technical and regulatory issues. A lack of standardized radiometric and spectral calibration workflows persists and affects consistent data processing. This also involves the flexible use of drones, as flight operation is limited to stable atmospheric conditions when a detailed spectral analysis is intended. Regulatory frameworks need to be optimized in mutual consideration of stakeholders and public interests to keep drone projects operable and safe. Exemption permits should be made possible at a reasonable additional expense to achieve greater area coverage (e.g., through flights above the maximum allowed altitude or BVLOS flights) and should not pose a barrier to innovation.

(5) Researchers should intensely focus on integrating traditional RS data from satellites and aircraft to enable larger-scale forest monitoring. It has been shown that there is a significant weakness in interoperability that needs to be addressed in future work to exploit complementary data with strong synergies (e.g., data fusion). Future research should focus on multisensor and multisource monitoring strategies and tackle interoperability issues by establishing standardized data infrastructures.

(6) Researchers rely on commercial software across the whole data pipeline, from data acquisition to the final analysis. This dependency comes at the expense of more flexible open-source solutions and can adversely affect knowledge and technology transfer among stakeholders.

In view of further decreasing costs of drone technology, hybrid VTOL fixed-wing platforms are conceivable for future FHM. These would allow an easier handling in spatially limited locations while flying more efficiently to cover larger areas. Larger combustion-engine-powered UAVs might close the gap between UAV and aerial surveys in the near future, according to costs and revisit times. Equipped with onboard D-GNSS technology, this could promote a further spread of fixed-wing models and thus provide a better cost-benefit ratio per hectare. With modern laser technology and thermal and hyperspectral cameras attached to the UAV, the spectral and structural range would be optimally exhausted for a more holistic UAV-based FHM approach.

**Author Contributions:** Conceptualization, S.E., J.D., H.-J.K., E.E., D.T. and T.S.; methodology, S.E.; software, S.E.; formal analysis, S.E.; investigation, S.E.; data curation, S.E.; writing—original draft preparation, S.E.; writing—review and editing, S.E., J.D., J.F., A.S., E.E., H.-J.K., D.T. and T.S.; visualization, S.E.; supervision, D.T. and T.S.; project administration, H.-J.K.; funding acquisition, H.-J.K. All authors have read and agreed to the published version of the manuscript.

**Funding:** This research is based on funding from a project grant provided by the Bavarian State Ministry for Food, Agriculture, and Forestry (E058).

**Data Availability Statement:** The data that support the findings of this study are available from the corresponding author, S.E., upon reasonable request.

**Acknowledgments:** The authors thank all anonymous reviewers that provided valuable comments.

**Conflicts of Interest:** The authors declare no conflict of interest.

## Abbreviations

The following abbreviations are used in this manuscript:

| | |
|---|---|
| ALS | airborne laser scanning |
| ANN | artificial neural network |
| BVLOS | beyond visual line of sight |
| CHM | canopy height model |
| CNN | convolutional neural network |
| D-GNSS | differential global navigation satellite system |
| DBH | diameter at breast height |
| DSM | digital surface model |
| DTM | digital terrain model |
| ESA | European Space Agency |
| EU | European Union |
| k-NN | k-nearest neighbor |
| FHM | forest health monitoring |
| GCP | ground control point |
| GNSS | global navigation satellite system |
| LiDAR | light detection and ranging |
| MLC | maximum likelihood classifier |
| MTOM | maximum take-off mass |
| NDVI | normalized difference vegetation index |
| NDRE | normalized difference red-edge index |
| NIR | near-infrared |
| nm | nanometer |
| OBIA | object-based image analysis |
| OA | overall accuracy |
| PPK | postprocessed kinematic |
| PRI | photochemical reflectance index |
| RF | random forest |
| RGB | red green blue |
| RS | remote sensing |
| RTK | real-time kinematic |
| SfM | structure from motion |
| SVM | support vector machine |
| UAS | unmanned/uncrewed aircraft system |
| UAV-RS | unmanned/uncrewed aerial vehicle remote sensing |
| UN | United Nations |
| UAV | unmanned/uncrewed aerial vehicle |
| VLOS | visual line of sight |
| VOC | volatile organic compound |
| VTOL | vertical take-off and landing |

## Appendix A

**Table A1.** Pests: reviewed articles regarding insects as the cause of tree stress.

| Authors | Study Area (Country) | UAV Type | Sensor | Geometric Correction | Tree Species Affected | Stressor |
|---|---|---|---|---|---|---|
| Lehmann et al. [88] | Germany | Multirotor | Multispectral | GCPs | Common oak (*Quercus robur*) | Oak splendor beetle (*Agrilus biguttatus*) |
| Näsi et al. [134] | Finland | Multirotor | Hyperspectral, RGB | Coregistration | Norway spruce (*Picea abies*) | European spruce bark beetle (*Ips typographus* L.) |
| Minarik and Langhammer [205] | Czech Republic | Multirotor | Multispectral | GCPs | Norway spruce (*Picea abies*) | European spruce bark beetle (*Ips typographus* L.) |
| Cardil et al. [206] | Spain | Multirotor | RGB | GCPs | Pine | Pine processionary moth (*Thaumetopoea pityocampa*) |
| Klein Hentz and Strager [114] | United States | Multirotor | RGB | GCPs | Not specified | Magicicada |
| Näsi et al. [135] | Finland | Multirotor | Hyperspectral, RGB | Coregistration | Norway spruce (*Picea abies*) | European spruce bark beetle (*Ips typographus* L.) |
| Otsu et al. [143] | Spain | Multirotor | RGB | Coregistration | Pine (*Pinus sylvestris, P. nigra*) | Pine processionary moth (*Thaumetopoea pityocampa*) |
| Röder et al. [152] | Germany | Multirotor | RGB | GCPs | Norway spruce (*Picea abies*) | European spruce bark beetle (*Ips typographus* L.) |
| Stoyanova et al. [207] | Bulgaria | Multirotor | RGB, multispectral | - | Norway spruce (*Picea abies*), Scots pine (*Pinus sylvestris*) | Bark beetle (*Ips typographus, I. acuminatus, I. sexdentatus*) |
| Zhang et al. [162] | China | Multirotor | Hyperspectral, RGB | Coregistration | Chinese pine (*Pinus tabulaeformis*) | Chinese pine caterpillar (*Dendrolimus tabulaeformis*) |
| Cardil et al. [129] | Spain | Multirotor | RGB, multispectral | - | Scots pine (*Pinus sylvestris*) | Pine processionary moth (*Thaumetopoea pityocampa*) |

**Table A1.** *Cont.*

| Authors | Study Area (Country) | UAV Type | Sensor | Geometric Correction | Tree Species Affected | Stressor |
|---|---|---|---|---|---|---|
| Dimitrov et al. [131] | Bulgaria | Fixed-Wing | Multispectral | - | Norway spruce (*Picea abies*), silver fir (*Abies alba*) | European spruce bark beetle (*Ips typographus* L.) |
| Klouček et al. [92] | Czech Republic | Multirotor | RGB, multispectral | GCPs | Norway spruce (*Picea abies*) | European spruce bark beetle (*Ips typographus* L.) |
| Lin et al. [116] | China | Not specified | Hyperspectral, LiDAR | GCPs | Yunnan Pine (*Pinus yunnanensis*) | Pine shoot beetle (*Tomicus* spp.) |
| Otsu et al. [138] | Spain | Not specified | RGB, multispectral | GCPs | Pine (*Pinus sylvestris, P. nigra*) | Pine processionary moth (*Thaumetopoea pityocampa*) |
| Akıncı and Göktoğan [125] | Turkey | Multirotor | RGB | - | Turkish pine (*Pinus brutia*) | Pine processionary moth (*Thaumetopoea pityocampa, T. wilkinsoni*) |
| Safonova et al. [208] | Russia | Multirotor | RGB | - | Siberian fir (*Abies sibirica*) | Four-eyed fir bark beetle (*Polygraphus proximus*) |
| Abdollahnejad and Panagiotidis [102] | Czech Republic | Multirotor | Multispectral | - | Norway spruce (*Picea abies*), Scots pine (*Pinus sylvestris*) | European spruce bark beetle (*Ips typographus* L.) |
| Duarte et al. [133] | Portugal | Fixed-Wing | Multispectral | GCPs | Eucalyptus (*Eucalyptus globulus*) | Eucalyptus longhorned borers (*Phoracantha semipunctata, P. recurva*) |
| Liu et al. [140] | China | Multirotor | Hyperspectral | Direct georeferencing (RTK/PPK), coregistration | Yunnan pine (*Pinus yunnanensis*) | Pine shoot beetle (*Tomicus* spp.) |
| Minařík et al. [120] | Czech Republic | Multirotor | Multispectral | GCPs | Norway spruce (*Picea abies*), Scots pine (*Pinus sylvestris*) | European spruce bark beetle (*Ips typographus* L.) |
| Barmpoutis et al. [209] | Greece | Multirotor | RGB | - | Pine (*Pinus brutia, P. halepensis, P. pinea*) | Pine shoot beetle (*Tomicus piniperda*) |

**Table A1.** *Cont.*

| Authors | Study Area (Country) | UAV Type | Sensor | Geometric Correction | Tree Species Affected | Stressor |
|---|---|---|---|---|---|---|
| Zhang et al. [210] | China | Multirotor | RGB, hyperspectral | Coregistration | Chinese pine (*Pinus tabulaeformis*) | Chinese pine caterpillar (*Dendrolimus tabulaeformis*) |
| Cessna et al. [96] | United States | Multirotor | RGB, multispectral | GCPs | Norway spruce (*Picea abies*) | Four-eyed spruce beetle (*Dendroctonus rufipennis*) |
| Lin et al. [117] | China | Not specified | Hyperspectral, LiDAR | GCPs | Yunnan Pine (*Pinus yunnanensis*) | Pine shoot beetle (*Tomicus* spp.) |
| Minařík et al. [130] | Czech Republic | Multirotor | Multispectral | Direct georeferencing (RTK/PPK) | Norway spruce (*Picea abies*) | European spruce bark beetle (*Ips typographus* L.) |
| Nguyen et al. [155] | Japan | Multirotor | RGB | - | Fir (*Abies mariesii*) | Tortrix moth (*Epinotia piceae*), bark beetle (*Polygraphus proximus*) |
| Paczkowski et al. [93] | Germany | Multirotor | Gas sensor | - | Norway spruce (*Picea abies*) | Bark beetle (*Ips typographus, Pityogenes chalcographus*) |
| Safonova et al. [87] | Bulgaria | Multirotor, Fixed-Wing | RGB, multispectral | - | Norway spruce (*Picea abies*) | European spruce bark beetle (*Ips typographus* L.) |
| Schaeffer et al. [110] | Mexico | Multirotor | RGB, multispectral | - | Piñon pine (*Pinus cembroides*) | Bark beetle (*Dendroctonus mexicanus*) |
| Koontz et al. [22] | United States | Multirotor | RGB, multispectral | GCPs | Ponderosa pine (*Pinus ponderosa*) | Western pine beetle (*Dendroctonus brevicomis*) |
| Pádua et al. [115] | Portugal | Fixed-Wing | Multispectral | GCPs | Sweet chestnut (*Castanea sativa*) | Oriental chestnut gall wasp (*Dryocosmus kuriphilus*), chestnut ink disease (*Phytophthora cinnamomi*), chestnut blight (*Cryphonectria parasitica*) |

**Table A1.** *Cont.*

| Authors | Study Area (Country) | UAV Type | Sensor | Geometric Correction | Tree Species Affected | Stressor |
|---|---|---|---|---|---|---|
| Kampen et al. [164] | Austria | Multirotor | Multispectral | - | Norway spruce (*Picea abies*), European ash (*Fraxinus excel- sior*) | European spruce bark beetle (*Ips typographus* L.), ash dieback disease (*Hymenoscyphus pseudoalbidus*) |
| Honkavaara et al. [94] | Finland | Multirotor | 2× RGB, multispectral, thermal, hyperspectral | Direct georeferencing (RTK/PPK), GCPs | Norway spruce (*Picea abies*) | European spruce bark beetle (*Ips typographus* L.), root and butt rot (*Heterobasidion annosum*) |

**Table A2.** Diseases: reviewed articles on diseases as the cause of tree stress.

| Authors | Study Area (Country) | UAV Type | Sensor | Geometric Correction | Tree Species affected | Stressor |
|---|---|---|---|---|---|---|
| Park and Kim [211] | South Korea | Fixed-Wing | RGB | - | Pine | Pine wilt disease (*Bursaphelenchus xylophilus*) |
| Smigaj et al. [136] | United Kingdom | Fixed-Wing | Thermal, multispectral | Coregistration | Pine (*Pinus sylvestris, P. contorta*) | Red band needle blight (*Dothistroma septosporum*) |
| Michez et al. [89] | Belgium | Fixed-Wing | RGB, multispectral | GCPs, coregistration | Black alder (*Alnus glutinosa*) | *Phytophthora alni* |
| Dash et al. [103] | New Zealand | Multirotor | Multispectral | GCPs | Monterey pine (*Pinus radiata*) | Herbicide |
| Brovkina et al. [90] | Czech Republic | Multirotor | RGB, multispectral | GCPs | Norway spruce (*Picea abies*) | Honey fungus (*Armillaria ostoyae*) |
| Dash et al. [104] | New Zealand | Multirotor | Multispectral | GCPs | Monterey pine (*Pinus radiata*) | Herbicide |
| Ganthaler et al. [122] | Austria | Multirotor | RGB | - | Norway spruce (*Picea abies*) | Needle bladder rust disease (*Chrysomyxa rhododendri*) |
| Gerard et al. [123] | United Kingdom | Multirotor | Hyperspectral | - | Common oak (*Quercus robur*) | Acute oak decline (AOD) |

**Table A2.** *Cont*.

| Authors | Study Area (Country) | UAV Type | Sensor | Geometric Correction | Tree Species Affected | Stressor |
|---------|---------------------|----------|--------|---------------------|----------------------|----------|
| Sandino et al. [163] | Australia | Multirotor | Hyperspectral | GCPs | Paperbark tea tree (*Melaleuca quinquenervia*) | Myrtle rust (*Austropuccinia psidii*) |
| Dell et al. [159] | Indonesia | Multirotor | RGB | GCPs | Eucalyptus (*Eucalyptus pellita*) | Bacterial wilt (*Ralstonia* sp.) |
| Jung and Park [212] | South Korea | Fixed-Wing | RGB, multispectral | - | Pine | Pine wilt disease (*Bursaphelenchus xylophilus*) |
| Lee et al. [107] | South Korea | Multirotor | RGB | - | Oak (*Quercus mongolica, Q. serrata, Q. dentate*) | Oak wilt (*Raffaelea quercus-mongolicae*) |
| Navarro et al. [158] | Portugal | Fixed-Wing | Multispectral | GCPs | Cork oak (*Quercus suber*) | Cork oak decline |
| Smigaj et al. [137] | United Kingdom | Multirotor | Thermal | Coregistration | Scots pine (*Pinus sylvestris*) | Red band needle blight (*Dothistroma septosporum*) |
| Deng et al. [213] | China | Fixed-Wing | RGB | - | Chinese red pine (*Pinus massoniana*) | Pine wilt disease (*Bursaphelenchus xylophilus*) |
| Iordache et al. [214] | Portugal | Multirotor | RGB, multispectral, hyperspectral | Direct georeferencing (RTK/PPK), GCPs | Maritime pine (*Pinus pinaster*) | Pine wilt disease (*Bursaphelenchus xylophilus*) |
| Syifa et al. [157] | South Korea | Multirotor | RGB | - | Pine (*Pinus densiflora, P. thunbergii*) | Pine wilt disease (*Bursaphelenchus xylophilus*) |
| Hoshikawa and Yamamoto [215] | Japan | Multirotor | RGB, multispectral | - | Japanese black pine (*Pinus thunbergii*) | Pine wilt disease (*Bursaphelenchus xylophilus*) |
| Tao et al. [216] | China | Multirotor | RGB | - | Chinese red pine (*Pinus massoniana*) | Pine wilt disease (*Bursaphelenchus xylophilus*) |
| Guerra-Hernández et al. [119] | Portugal | Fixed-Wing | RGB, multispectral | GCPs | Black alder (*Alnus glutinosa*) | *Phytophthora alni* |

**Table A2.** *Cont.*

| Authors | Study Area (Country) | UAV Type | Sensor | Geometric Correction | Tree Species Affected | Stressor |
|---|---|---|---|---|---|---|
| Li et al. [124] | China | Multirotor | RGB | - | Chinese pine (*Pinus tabulaeformis*) | Pine wilt disease (*Bursaphelenchus xylophilus*) |
| Qin et al. [141] | China | Multirotor | Multispectral | Direct georeferencing (RTK/PPK) | Chinese red pine (*Pinus massoniana*) | Pine wilt disease (*Bursaphelenchus xylophilus*) |
| Sun et al. [217] | China | Multirotor | RGB | - | Chinese red pine (*Pinus massoniana*) | Pine wilt disease (*Bursaphelenchus xylophilus*) |
| Wu et al. [111] | China | Mulirotor | RGB | - | Chinese pine (*Pinus tabulaeformis*) | Pine wilt disease (*Bursaphelenchus xylophilus*) |
| Xia et al. [44] | China | Fixed-Wing | RGB | - | Pine (*Pinus thunbergii, P. densiflora*) | Pine wilt disease (*Bursaphelenchus xylophilus*) |
| Yu et al. [112] | China | Multirotor | RGB, Hyperspectral, LiDAR | GCPs | Korean pine (*Pinus koraiensis*) | Pine wilt disease (*Bursaphelenchus xylophilus*) |
| Yu et al. [118] | China | Multirotor | Multispectral, RGB | Direct georeferencing (RTK/PPK) | Chinese red pine (*Pinus massoniana*) | Pine wilt disease (*Bursaphelenchus xylophilus*) |
| Yu et al. [142] | China | Multirotor | Hyperspectral, LiDAR | GCPs | Chinese red pine (*Pinus massoniana*) | Pine wilt disease (*Bursaphelenchus xylophilus*) |
| Yu et al. [146] | China | Multirotor | Hyperspectral | GCPs | Chinese pine (*Pinus tabulaeformis*) | Pine wilt disease (*Bursaphelenchus xylophilus*) |

**Table A2.** *Cont.*

| Authors | Study Area (Country) | UAV Type | Sensor | Geometric Correction | Tree Species Affected | Stressor |
|---------|---------------------|----------|--------|---------------------|----------------------|----------|
| Pádua et al. [109] | Portugal | Multirotor | RGB, multispectral | GCPs | Sweet chestnut (*Castanea sativa*) | Chestnut ink disease (*Phytophthora cinnamomi*), chestnut blight (*Cryphonectria parasitica*), nutritional deficiencies |
| Pádua et al. [115] | Portugal | Fixed-Wing | Multispectral | GCPs | Sweet chestnut (*Castanea sativa*) | Chestnut ink disease (*Phytophthora cinnamomi*), chestnut blight (*Cryphonectria parasitica*), Oriental chestnut gall wasp (*Dryocosmus kuriphilus*) |
| Kampen et al. [164] | Austria | Multirotor | Multispectral | - | Norway spruce (*Picea abies*), European ash (*Fraxinus excelsior*) | Ash dieback disease (*Hymenoscyphus pseudoalbidus*), European spruce bark beetle (*Ips typographus L.*) |
| Honkavaara et al. [94] | Finland | Multirotor | 2× RGB, multispectral, thermal, hyperspectral | Direct georeferencing (RTK/PPK), GCPs | Norway spruce (*Picea abies*) | Root and butt rot (*Heterobasidion annosum*), European spruce bark beetle (*Ips typographus L.*) |

**Table A3.** Phytoparasites: reviewed articles regarding phytoparasites as a cause of tree stress.

| Authors | Study Area (Country) | UAV Type | Sensor | Geometric Correction | Tree Species Affected | Stressor |
|---------|---------------------|----------|--------|---------------------|----------------------|----------|
| Li et al. [144] | Costa Rica | Multirotor | Multispectral | GCPs | Tropical dry forest | Lianas |
| Maes et al. [108] | Australia | Multirotor | RGB, thermal | GCPs | Eucalyptus (*Eucalyptus fibrosa, E. moluccana*) | Box mistletoe (*Amyema miquelii*) |
| Waite et al. [145] | Malaysia | Multirotor | RGB | - | Tropical forest | Liana |
| Yuan et al. [166] | Costa Rica | Multirotor | Multispectral, thermal | GCPs | Neotropical dry forest | Liana |
| Miraki et al. [113] | Iran | Multirotor | RGB | GCPs | Persian ironwood (*Parrotia persica*) | Common mistletoe (*Viscum album*) |

**Table A4.** Abiotic stressors: reviewed articles regarding abiotic agents of tree stress.

| Authors | Study Area (Country) | UAV Type | Sensor | Geometric Correction | Tree Species Affected | Stressor |
|---|---|---|---|---|---|---|
| Hernández-Clemente et al. [85] | Spain | Fixed-Wing | Multispectral | - | Scots pine (*Pinus sylvestris*) | Drought |
| Fraser et al. [149] | Canada | Multirotor | RGB | - | Jack pine (*Pinus banksiana*), spruce (*Picea glauca, P. mariana*) | Fire |
| Ludovisi et al. [147] | Italy | Multirotor | Thermal | GCPs | Black poplar (*Populus nigra*) | Drought |
| McKenna et al. [101] | Australia | Fixed-Wing | RGB | GCPs | Eucalypt forest | Fire |
| Buras et al. [150] | Germany | Multirotor | Multispectral | - | Scots pine (*Pinus sylvestris*) | Drought |
| Nagai et al. [169] | Japan | Multirotor | RGB | - | Japanese cedar (*Cryptomeria japonica*) | Heavy snow |
| Rossi et al. [91] | Argentina | Fixed-Wing | Multispectral | Direct georeferencing (GNSS) | Subtropical forest | Fire |
| Arkin et al. [154] | Canada | Multirotor | RGB | GCPs | Douglas-fir (*Pseudotsuga menziesii*), white spruce (*Picea engelmannii × Picea glauca*), lodgepole pine (*Pinus contorta*) | Fire |
| Carvajal-Ramírez et al. [99] | Spain | Multirotor | RGB, multispectral | Direct georeferencing (RTK/PPK), GCPs | Mediterranean forest | Fire |
| Padua et al. [86] | Portugal | Multirotor, Fixed-Wing | RGB, mulispectral | - | Maritime pine (*Pinus pinaster*) | Fire |
| Rossi and Becker [161] | Argentina | Fixed-Wing | RGB | - | Subtropical forest | Fire |
| Shin et al. [218] | South Korea | Multirotor | Multispectral | - | Korean red pine (*Pinus densiflora*) | Fire |
| Zhu et al. [139] | China | Multirotor | RGB, LiDAR | Direct georeferencing (RTK/PPK) | Mangrove (*Kandelia obovate, Avicennia marina, Aegiceras corniculatum*) | Inundation stress |
| Campbell et al. [153] | United States | Multirotor | RGB | GCPs | Piñon woodland | Drought |

**Table A4.** *Cont*.

| Authors | Study Area (Country) | UAV Type | Sensor | Geometric Correction | Tree Species Affected | Stressor |
|---|---|---|---|---|---|---|
| Pádua et al. [151] | Portugal | Fixed-Wing | RGB, multispectral | - | Maritime pine (*Pinus pinaster*) | Fire |
| Talucci et al. [132] | Russia | Multirotor | RGB, multispectral | GCPs | Cajander larch forest (*Larix cajanderi*) | Fire |
| Tran et al. [219] | South Korea | Multirotor | RGB | - | Not specified | Fire |
| Viedma et al. [95] | Spain | Not specified | LiDAR | Direct georeferencing (RTK/PPK) | Aleppo pine (*Pinus halepensis*) | Fire |
| Araujo et al. [97] | Panama | Not specified | RGB | - | Tropical forest | Extreme rainfall |
| Cohen et al. [98] | United States | Multirotor | RGB | GCPs | Black mangrove (*Avicennia germinans*) | Winter freeze |
| D'Odorico et al. [105] | Switzerland | Multirotor | Multispectral (modified to sense PRI) | GCPs | Scots pine (*Pinus sylvestris*) | Drought |
| Hillman et al. [100] | Australia | Multirotor | RGB, LiDAR | Direct georeferencing (RTK/PPK), GCPs, Coregistration | Eucalyptus (*Eucalyptus obliqua, E. globulus*) | Fire |
| Woo et al. [220] | South Korea | Multirotor | Multispectral | - | Korean red pine (*Pinus densiflora*), Japanese larch (*Larix kaempferi*) | Fire |
| Pádua et al. [109] | Portugal | Multirotor | RGB, multispectral | GCPs | Sweet chestnut (*Castanea sativa*) | Nutritional deficiencies, chestnut ink disease (*Phytophthora cinnamomi*), chestnut blight (*Cryphonectria parasitica*) |
| Koontz et al. [22] | United States | Multirotor | RGB, multispectral | GCPs | Ponderosa pine (*Pinus ponderosa*) | Drought, Western pine beetle (*Dendroctonus brevicomis*) |

**Table A5.** Stressor not defined/unknown: reviewed articles in which no stressor was defined.

| Authors | Study Area (Country) | UAV Type | Sensor | Geometric Correction | Tree Species Affected | Stressor |
|---|---|---|---|---|---|---|
| Yuan and Hu [126] | China | - | RGB | - | Not specified | Not specified/unknown |
| Saarinen et al. [34] | Finland | Single-rotor helicopter | RGB, hyperspectral | GCPs | Norway spruce (*Picea abies*), Scots pine (*Pinus sylvestris*), downy birch (*Betula pendula and B. pubescens*) | Not specified/unknown |
| Barmpoutis et al. [121] | Greece | Multirotor | RGB | - | Fir | Not specified/unknown |
| Khokthong et al. [148] | Indonesia | Multirotor | RGB | GCPs | Oil palm (*Elaeis guineensis*) | Not specified/unknown |
| Abdalla et al. [160] | Indonesia | Multirotor | RGB | - | Acacia | Not specified/unknown |
| Campos-Vargas et al. [156] | Costa Rica | Multirotor | Multispectral | GCPs | Tropical dry forest | Not specified/unknown |
| Gallardo-Salazar and Pompa-García [221] | Mexico | Multirotor | Multispectral | - | Arizona pine (*Pinus arizonica*) | Not specified/unknown |

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
