# Peer review of "UAV-Based Forest Health Monitoring: A Systematic Review"

_remotesensing, doi:10.3390/rs14133205_

Round 1

Reviewer 1 Report

Dear Authors,

   Many thanks for your manuscript submission to MDPI Journal of Remote Sensing. This research article presents a literature review on the past and current technical advances on unmanned aerial vehicle (UAV) based forest health monitoring (FHM) in the past decade, from simplest schemes to the recent machine learning based approaches which are threatened by biotic and abiotic stressors. The authors claimed that the survey covers the major advantages and generalized limitations (shortcomings) of UAV based techniques for FHM. Their study covered concrete investigations on how the current advances have been done and up to overcome the bottleneck issues. It is a strong survey paper, containing abundant technical details, presenting clearly classified techniques under different set of scenarios. 

   While it is a relatively long survey paper, the use of English is also quite good, the structure of paper is also complete and meticulous. Most of the figures and tables are decently arranged. Therefore, I will recommend this version as "Accept with minor revisions" to Editorial Office, after applying the related edits along with other reviewers' suggestions. There are a few aspects requiring the authors' attention when editing this research article. Major problematic issues can be specified as follows (may not limit to all of these listed below):

  a) Abstract: The abstract session is a bit too long, it would be acceptable on lacking parallel comparison to the existing review papers, while it also skipped the merits of advantages on UAV. Besides, I think it had missed to present a few short sentences, summarizing keynote conclusion of your study. Also, the number of index terms is 10 (typically 4-6 for research article), please check the upper limit of survey paper requirement.

   b) Introduction: I may recommend the authors slightly adjust and optimize the organization of these subsections therein, and present a summary of major contributions of your work in very clear scenarios. Regarding the last paragraph, I would recommend to state "The organization on the remainder of this paper is as follows: ...", which should look even better. 

   c) Section 2: Materials and Methods: The context of this section should be expanded. The current version has two subsections (which should also be extended), but the top figure (Fig. 5) seems to be in absence with some statement in correspondence. Besides, the authors may skipped to present some related work (literature research is a bit too generic), and I think the dataset structuring is also not specific. Please consider applying further edits towards this section. 

   d) Figures and Tables: Some figures should be updated: the low resolution and distortion issues persist in Figs. 1, 5, and 6. The twisted characters in Fig. 1, the tiny digits at Fig. 6, and the uniform font style problem within these figures should be corrected. The authors may also need to adjust the size and resolution of figures to comply with MDPI publication template.

The authors presented all the tables in Appendix A, I think the linspacing issues of each table should be calibrated. They are a bit too loose, and it is better not crossing over pages (except for Table 2).

   e) Results: this section is basically fine, I think if furthter quantitative results can be tabulated, it may help understanding the following topics. Besides, a few subsections need minor adjustment, a few trivial issues also need to be fixed in the proofreading process.

   f) Discussions: This section presents very good observations on recent advances of most recent UAV-based techniques along with their limitations. I think for each manifold of limitations, the authors may emphasize the tree structure on each aspect of their concerns. It could be better to initialize subsections similar as prior sections. Parallel comparison should be clear.

   g) Conclusion section: I suggest on expanding the last paragraph to the following scenarios: future investigation in accordance with their opening questions, prospective work with respect to limitations of study, prospective research orientations with respect to the future study. In general this section is very good, while the last paragraph needs some post-processing, which enable balanced demands and qualify your expected work for publication.

   h) References: while it has already contained more than 200 cited journals and conferences, there are a couple of minor issues to be fixed. For instance, it is unnecessary to capitalize all letters on the title of conference proceeding paper at Ref. [200]. Some redundant half-space blanks persist in some of the cited work. The citation style of a few online source based work should be double-checked to ensure obedience with MDPI template. Please update and thanks for your careful edits in advance.  

   Minor problematic issues suggested for your improvements:

   1) I think in the list of abbreviations at Page 27 of 47, the authors need to check if some terms have the real, exact expressions, i.e., kNN, OA, UAV-RS. Also, the authors need to check if the used abbreviation terms may be confused with others, i.e., UAV. 

   2) Consider condensing some of the tabless which are too long while less important to the essence of your survey journal. 

   3) There are some redundant spaces in the context and in the Reference, i.e., one at Line 305, 325, 437 1164, 1207, etc, which should be eliminated. The r and p at lines 659-660 should be italic. Also, be sure the length of your survey paper does not exceed the page limit of MDPI Journal (Remote Sensing).

   4) Overall, the writing skills as displayed in the current version of paper looks fairly good, in some specific subsections, calibrations on grammatical issues should be performed, and some phrases must be revised. 

  Thank you very much for the diligent work, we look foward to seeing your polished manuscript coming into final acceptance. Good luck!

Best wishes,

Yours sincerely,  

Reviewer 2 Report

the manuscript presented a comprehensive descriptions of UAVs for forest monitoring. As a review rather than an article, all figures should be well improved. This should be well revised before final acceptance. 

I recommended some reference for this field.

radiometric calibration:

Radiometric calibration for multispectral camera of different imaging conditions mounted on a UAV platform

Radiometric calibration of UAV remote sensing image with spectral angle constraint

multispectral images:

Machine Learning-Based Approaches for Predicting SPAD Values of Maize Using Multi-Spectral Images

Bagaram, Martin B., et al. "UAV remote sensing for biodiversity monitoring: are forest canopy gaps good covariates?." Remote Sensing 10.9 (2018): 1397.

Torresan, Chiara, et al. "Forestry applications of UAVs in Europe: A review." International Journal of Remote Sensing 38.8-10 (2017): 2427-2447.    

Yuan, Chi, Youmin Zhang, and Zhixiang Liu. "A survey on technologies for automatic forest fire monitoring, detection, and fighting using unmanned aerial vehicles and remote sensing techniques." Canadian journal of forest research 45.7 (2015): 783-792.    

Salamí, Esther, Cristina Barrado, and Enric Pastor. "UAV flight experiments applied to the remote sensing of vegetated areas." Remote Sensing 6.11 (2014): 11051-11081.    

Adão, Telmo, et al. "Hyperspectral imaging: A review on UAV-based sensors, data processing and applications for agriculture and forestry." Remote sensing 9.11 (2017): 1110. 

Reviewer 3 Report

The paper reviews papers about employing UAVs for Forest Health Monitoring.

The paper is very interesting, and the author conducted a very comprehensive study. The research methodology, as the most important section of a review paper, is robust; however, some minor comments should be addressed:

Why only WOS has been used, while some sources are not covered by it and in most review papers scopus beside WOS is employed.

In addition why the studies before 20112 have not been condiderd in this study.

Figure 1 is a bit unclear; its presentation should be improved.

The abstract I unnecessary too long

In keywords, both Unmanned aerial vehicles and UAVs have been used. Please avoid using acronyms in keywords.

On pages 17, 18 , and 19 it seems some headings do not have a numbering system.

Some sections, particularly the introduction, are excessively lengthy and should be condensed.
